# Enhanced neural plasticity in monkey TE compared to TEO during learning of a feature-ambiguous visual categorization task

Wenliang Wang [1,4], Bing Li [1,4], Narihisa Matsumoto[2,4], Kariely Martinez Gomez[1], Kazuko Hayashi[2], Yasuko Sugase-Miyamoto [2], Richard C. Saunders[1], Barry J. Richmond [1] ✉ & Mark A. G. Eldridge [1,3] ✉

Primates can rapidly categorize images by visual feature similarity. We previously showed that inferior temporal cortex (IT) subregions TEO and TE contribute to visual categorization to differing extents. To investigate the neural plasticity underlying visual categorization, we recorded simultaneously from both areas while two male monkeys learned a visual categorization task. Category specificity and generalization are initially stronger in TEO than in TE but increase with learning only in the TE neuronal population, whose neural representations correlate with behavioral performance. TE and TEO can contribute complementary, partially independent category information. However, TEO does not add learning-relevant variance across days. TE exhibits greater categorical enhancement for correct than error trials compared with TEO. Single-neuron analyses revealed that TE category selectivity strengthens with learning, primarily through enhanced encoding of one category. Combined with lesion evidence, our results suggest that TE plasticity likely plays a more fundamental role in supporting visual category learning.

Visual categorization is remarkably rapid and accurate in primates and is critical for supporting quick behavioral responses to desirable or threatening visual stimuli. Categorization is thought to be processed, at least partially, within inferior temporal cortex (IT)[1,2]. In nonhuman primates, IT consists of two subregions, TE and TEO, distinguished by their architectonics and connectivity[3–6]. These distinctions are further supported by functional studies[7] and more recent fMRI investigations detailing retinotopic organization[8]. Anatomically and functionally, TEO is one step earlier than TE in the canonical hierarchical ventral visual stream. Deficits in visual categorization are caused by lesions to TE and/or TEO[9–12], and by microstimulation of TE[13]. Our recent study

showed that bilateral lesions of TE or TEO caused mild impairments in categorization whereas combined TE and TEO removal was followed by a severe, long-lasting impairment[12]. This observation is inconsistent with a simple, serial feed-forward ventral visual hierarchy, but consistent with the "recurrent occipitotemporal network" proposal[14]. We recently proposed that, in visual categorization, TE and TEO work in parallel[12,15].

Both TE and TEO contain single neurons with responses selective for visual categories such as faces, bodies, and scenes[16–21]. But the study of category representation in the activity of neuronal populations has been restricted mainly to TE[22–26]. The evaluation of neural plasticity

[1]Laboratory of Neuropsychology, National Institute of Mental Health, National Institutes of Health, Bethesda, MD, USA. [2]Human Informatics and Interaction Research Institute, National Institute of Advanced Industrial Science and Technology (AIST), Tsukuba, Ibaraki, Japan. [3]Present address: Biosciences Institute, Newcastle University, Newcastle upon Tyne, UK. [4]These authors contributed equally: Wenliang Wang, Bing Li, Narihisa Matsumoto.
✉e-mail: barryrichmond@mail.nih.gov; mark.eldridge@newcastle.ac.uk

underlying visual categorization has, likewise, been restricted to TE. Categorization or discrimination learning can shape the responses of TE neurons, e.g., enhance the representation of task-related visual features[27–30] and category information[30,31].

Our lesion studies suggest that there is parallel processing between TE and TEO during categorization. To assess the potential for parallel processing of visual categorization in these two areas at the neuronal scale, we measured population activity in TE and TEO simultaneously, while two monkeys were learning a visual categorization task.

## Results

We trained two monkeys (T and X) to categorize morphed images as either "dog-like "or "cat-like" in a visually cued two-interval forced choice (2-IFC) paradigm[11] (Fig. 1A). In each trial, one image was pseudo-randomly selected from a set of morphed images (Fig. 1B, monkey T: 10 morph identities × 11 morph levels; monkey X: 20 morph identities × 11 morph levels. Morph identity refers to one specific morph sequence with a unique pair of dog-cat prototypes). To receive liquid reward,

monkey T was required to release the bar in the first interval signaled by a red square for "cat-like" images (morph level: 0%, 25%, 35%, 40%, 45%) or release the bar in the second interval signaled by a green square for "dog-like" images (morph level: 55%, 60%, 65%, 75%, 100%). Otherwise, the monkey would receive a time-out (4 s) as punishment for the incorrect choices. To balance the subjective value of outcomes in cat and dog trials, a larger reward was delivered in correct dog trials than in cat trials to account for the delay discounting effect (Fig. 1A, also see details in "Methods"). Monkey X performed a variation of this task, in which it could avoid a time-out (4 s) and get a liquid reward by releasing the bar in the second interval for "dog-like" images. A release in the first interval always led to a new trial−irrespective of the category of image presented−without a time-out or reward. Both monkeys received a time-out or a reward randomly for 50% morphed images.

For both monkeys, behavioral performance improved across days (Figs. 1C and S1A). Reaction times for correct bar-releases in the first interval for cat-like images decreased across days (Fig. S1B). Potential choice biases due to the unbalanced 2-IFC were evaluated by monkeys'

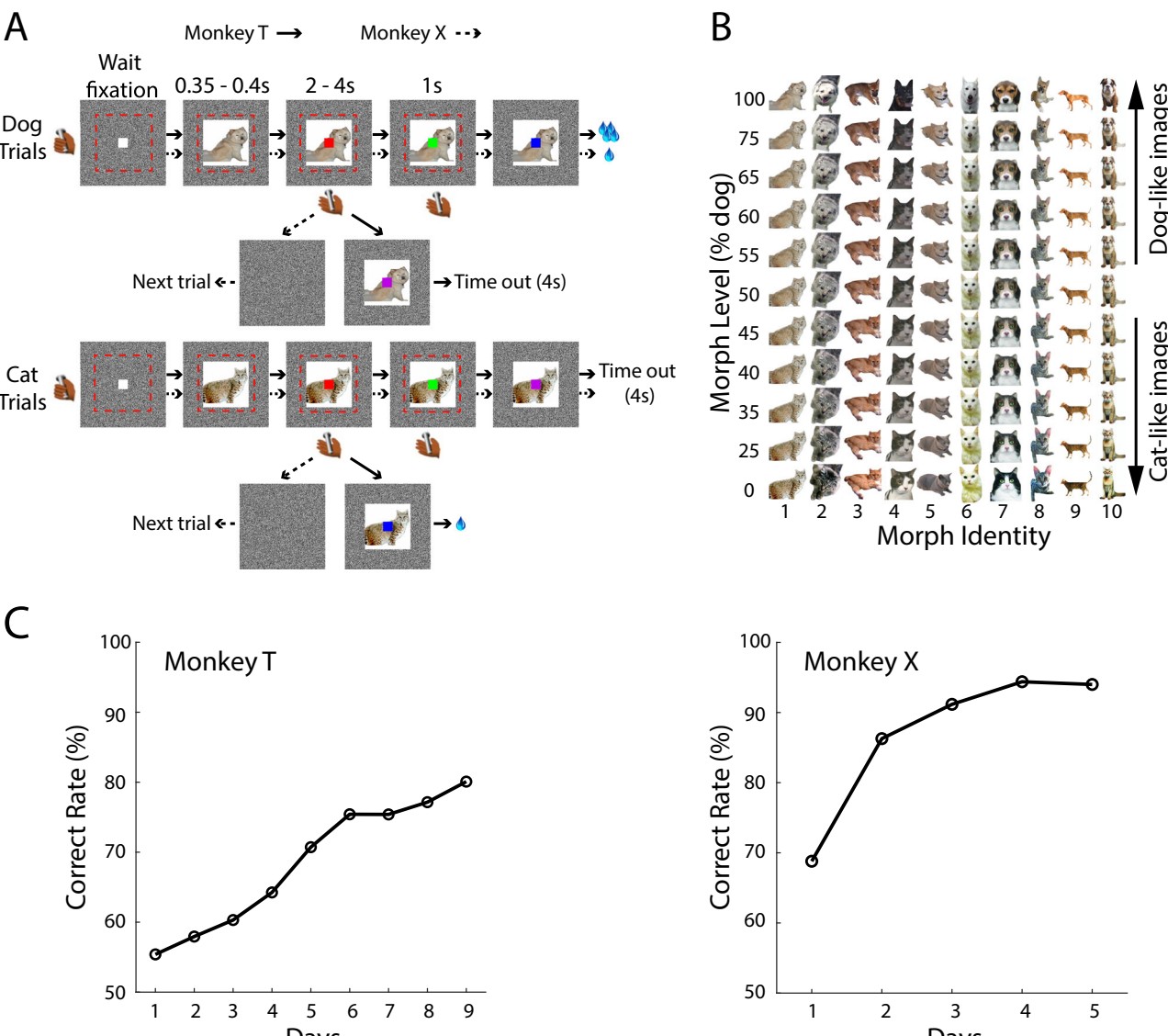

**Fig. 1 | Task design and behavior. A** Sequence of events in individual dog and cat trials for monkey T (solid arrows) and X (dashed arrows). Cat/dog images designed by Freepik (www.freepik.com). **B** Matrix of cat−dog morphed images used as visual stimuli for monkey T. For monkey X, 20 morph identity were used. Cat/dog prototype images designed by Freepik (www.freepik.com). **C** Correct rate of monkeys' choice on each learning day. Performance on images at 50% morph level was not included. T: $p = 5.77 \times 10^{-6}$, $r = 0.98$; X: $p = 0.055$, $r = 0.87$, Pearson correlation, correct rate vs. day number.

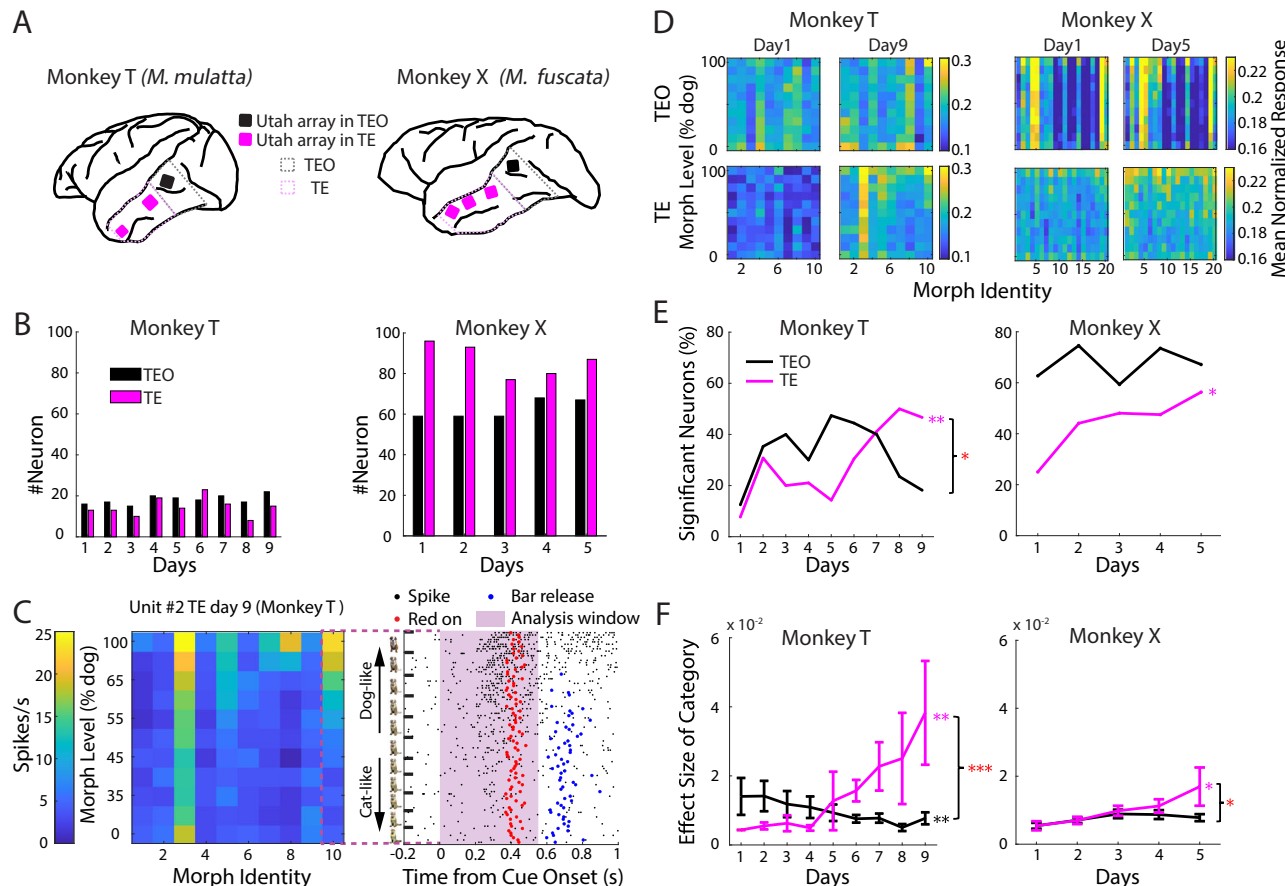

**Fig. 2 | Neurons in TE, not TEO, were increasingly modulated by category.**
**A** Locations of Utah arrays in TEO (black) and TE (magenta). Large/small squares = 96/64-channel arrays. **B** Number of units recorded per day in TE and TEO. **C** Example responses to the image matrix. Left: color-coded responses of an example neuron. Right: spike raster (black dots), onset time of red square (red dots) and bar-release time (blue dots) in the first interval when images in 10th morph identity were presented. Time 0 corresponds to image onset. Events in the second interval are not shown. Trials were grouped by images. Cat/dog prototype images designed by Freepik (www.freepik.com). **D** Mean normalized population responses of neuron ensembles on the first vs. last day. **E** Percentage of neurons significantly modulated by visual categories in TE and TEO. T: $p_{(percentage-vs.-day)}$ = 0.99 (TEO), 0.0066 (TE) $r$ = 0.007, 0.82. X: $p_{(percentage-vs.-day)}$ = 0.76, 0.039, $r$ = 0.19, 0.9, Pearson correlation. Lower bounds of one-tailed 95% confidence interval (CI) of

$r_{(TE-vs.-day)} - r_{(TEO-vs.-day)}$: 0.05 (T), 0 (X), bootstrapping Pearson correlation (5000 bootstrap samples). Magenta (black) asterisks along the curves indicate a significant correlation between the proportion and day number for TE (TEO). Red asterisks by the brace indicate the proportion-vs.-day correlation are significantly stronger in TE than in TEO. **F** Effect size of visual category increased across learning for TE, not TEO category-modulated neurons. Red asterisks by the brace indicate significant interactions between brain region and day number. Notations are same across related figures. T: $p_{(effect-size-vs.-day)}$ = 0.0072 (TEO), 0.0025 (TE), $\beta_1$ = −0.11, 0.46; $n$(TEO) = 2, 6, 6, 6, 9, 8, 8, 4, 4 neurons (day 1–9); $n$(TE) = 1, 4, 2, 4, 2, 7, 6, 4, 7; X: $p_{(effect-size-vs.-day)}$ = 0.08 (TEO), 0.012 (TE), $\beta_1$ = 0.061, 0.29, LM (linear model, two-sided, same across figures). $n$(TEO) = 37, 44, 35, 50, 45 neurons (day 1–5); $n$(TE) = 24, 41, 37, 38, 49. $p_{(region×day)}$ = 3.2 × 10⁻⁵ (T), 0.044 (X), LM. All data are represented as mean ± SEM. *$p$ < 0.05, **$p$ < 0.01, ***$p$ < 0.001.

performance on 50% morphed images. For monkey X, significant choice biases to "dog-like" images were observed in the first three learning days, presumably because monkey X could only obtain a reward in dog trials. Choice biases were not observed in the following 2 days. Across learning, choice bias was significantly reduced in monkey X (Fig. S1C). For monkey T, no significant bias was observed on any learning days (Fig. S1C), which could be attributed to the balanced subjective value associated with "cat-like" and "dog-like" images. Despite their differing decision biases, both monkeys displayed systematic increases in psychometric slope over training (Fig. S1D), indicating enhanced category sensitivity.

**Neural activity in TE, not TEO, is modulated by visual category learning**

Neural activity was recorded simultaneously in TE and TEO with implanted 64- or 96-channel Utah arrays (Fig. 2A). We recorded 164 neurons in TEO and 131 neurons in TE during learning (day 1–9) from monkey T, and 312 neurons in TEO and 433 neurons in TE (day 1–5) from monkey X (Fig. 2B).

We focused on the image presentation period before the earliest bar-release in the first interval (Figs. 2C and S1E. T: 0 - 550 ms; X: 0 - 530 ms). Example neurons, from both TE and TEO, showed different tuning to category, morph level and morph identity (Figs. 2C and S2). Mean normalized responses of the TE neuron population, but not that of TEO, showed greater separation between responses to dog-like and cat-like images on the last day compared to the first (Fig. 2D). This increased separation was primarily driven by an overall enhancement of responses to dog-like images, as detailed in the following sections.

To test neural correlates of category learning in TE and TEO, a nested ANOVA was carried out for each neuron on each day, with category, morph level (nested within category), and morph identity as factors. Responses to 50% morphed images were not included in the ANOVA. In TE, not TEO, the proportion of neurons significantly modulated by visual category significantly increased during learning (Fig. 2E. In monkey T, the proportion-vs.-day correlation was significantly stronger in TE than in TEO, whereas in monkey X the same trend did not reach significance).

A linear model (LM) showed that the effect size of visual category on TE, not TEO, neural activity significantly increased across learning days (Fig. 2F). There was no significant change in TEO responses for the other two factors—morph level or morph identity. For TE neurons, the effects of morph level and morph identity were not consistent between the two monkeys (Fig. S3).

Taken together, these analyses show that TE's category signal strengthened with learning in both monkeys: the TE-vs.-TEO difference is robust by effect-size in both monkeys and additionally confirmed by the percentage-of-significant-neurons metric in monkey T.

### Both category specificity and generalization increased during learning in TE populations, but not those in TEO

To evaluate the category representation in the neural population, we first trained a traditional category decoder (see Fig. 3A for a simplified illustration, also see "Methods"). Decoding accuracies of both TE and TEO populations were significantly above chance levels in almost all of the learning days (Fig. 3B). In the late days of learning, decoding accuracy of the TE population was significantly higher than TEO (Fig. 3B). Furthermore, consistent with the analysis at the single neuron level, decoding accuracy during learning increased monotonically for the TE neuronal population in both monkeys, but not for TEO population (Figs. 3B and S4A).

A good candidate brain area for categorization requires not only category specificity—i.e. to distinguish between exemplars of different categories, but also generalization, i.e. to categorize never-before-seen images, to generalize across exemplars of the same category[1]. However, high scores in the traditional decoding only guarantee category specificity, not generalization. We tested how well the neural populations generalize category identity by training a generalization category decoder with a 10-fold cross-validation strategy over morph identities (see Fig. 3D for a simplified illustration, also see "Methods"). At test, the decoder was challenged with never-before-seen images from the left-out morph identities (i.e. entire cat-dog morph sequences). Thus, high scores from this decoder required the existence of common response patterns that generalize from one morph identity to other morph identities within a category. We found similar results to the category decoder. The generalization of categorization performance in the TE, but not TEO, population increased monotonically across learning days (Figs. 3E and S4B).

Our lesion studies suggest that TE and TEO may function in parallel during categorization. To evaluate this, we computed a discriminability index (d′) from decoder scores for TE and TEO ensembles individually and in combination. Compared to decoding accuracy, d′ quantifies the separation between category distributions relative to their variability, offering a more sensitive and non-saturating measure of neural discriminability. Each region's d′ profile mirrored its decoding-accuracy pattern (Fig. 3C, F vs. Fig. 3B, E, $p_{(metric \times day)} > 0.05$ for all comparisons, LM). In both monkeys, and under both decoding strategies, d′ (TE + TEO) exceeded the stronger individual region—d′ (TE) or d′ (TEO)—on several days (Monkey T: 3/9 and 3/9 days; Monkey X: 4/5 and 2/5 days for category and generalization category decoders, respectively), yet remained below the sum of the two regions, i.e., d′ (TE) + d′ (TEO) (Fig. 3C, F). This pattern indicates that TE and TEO provide complementary, though partially overlapping, information that can enhance category representation when combined. On the remaining days, d′ (TE + TEO) was either equal to or less than the stronger region. These results indicate that TE and TEO can contribute to categorization in a complementary manner, but this benefit is not consistently expressed—potentially due to day-to-day variability in neural signals, and/or limited neuronal sampling (particularly in monkey T).

We further evaluated the category specificity and generalization of the neural populations by directly measuring the representational dissimilarity[25] (1 − r, Pearson correlation) between pairs of images from different categories or the same category, respectively. To be comparable with the generalization decoding strategy above, two images were taken from two different morph identities. Selection of image pairs in this way avoided the confound of visual perceptual similarity for the images within one morph identity. We found that the dissimilarity of TE population response patterns elicited by images from different categories significantly increased during learning for both monkeys (Fig. 4A, cat vs. dog, magenta lines). In contrast, the TE population response pattern elicited by images from the same category became increasingly similar (cat for monkey T; both cat and dog for monkey X. Fig. 4B, C). However, no significant changes in population response pattern were found in TEO (Fig. 4, black lines). Thus, both specificity and generalization of category representation increase in TE, but not TEO, during categorization learning.

Notably, category representation was initially stronger in TEO than in TE (Figs. 3B, C, E (inset for monkey X), F and 4A). As learning progressed, representation in TE surpassed that in TEO. This temporal crossover suggests that TEO may support early, coarse category representation, while TE undergoes experience-dependent plasticity to refine the encoding of well-learned categories (see "Discussion").

### Behavioral performance correlated with neural representations in TE but not TEO

To assess the relationship between neural and behavioral changes over learning, we used three approaches. First, we performed a shuffled control analysis by comparing the correlations between decoding accuracy and behavioral performance against correlations obtained using shuffled decoding accuracy, with a bootstrapping test to assess significance. Decoding accuracy of the TE population during learning correlated significantly with monkeys' performance, but that of the TEO population did not (Category decoder: TE: for monkey T, 95% CI of $r_{(TE)} - r_{(shuffle)}$: [0.04 0.82]; for monkey X, 99% CI of $r_{(TE)} - r_{(shuffle)}$: [0.3 1.86]. TEO: for monkey T, 95% CI of $r_{(TEO)} - r_{(shuffle)}$: [−0.68 0.46]; for monkey X, 95% CI of $r_{(TEO)} - r_{(shuffle)}$: [−0.02 1.67]; Generalization category decoder: TE: for monkey T, 99% CI of $r_{(TE)} - r_{(shuffle)}$: [0.04 1.37]; for monkey X, 95% CI of $r_{(TE)} - r_{(shuffle)}$: [0.16 1.67]. TEO: for monkey T, 95% CI of $r_{(TEO)} - r_{(shuffle)}$: [−0.9 0.08]; for monkey X, 95% CI of $r_{(TEO)} - r_{(shuffle)}$: [−0.31 1.01]).

Second, we applied a linear regression model directly comparing average decoding accuracy in area TE or TEO with behavioral performance across days. Given the limited statistical power in this approach when analyzing individual animals, we combined data from both monkeys. Decoding accuracy in TE, not TEO, was a significant predictor of behavior for both decoding strategies (Category decoder: $p_{(TE)} = 0.001$, $p_{(TEO)} = 0.41$. Generalization category decoder: $p_{(TE)} = 0.0006$, $p_{(TEO)} = 0.38$). This relationship did not depend on individual subject differences (Category decoder TE: $p_{(decoding\_accuracy \times monkey)} = 0.47$, TEO: $p_{(decoding\_accuracy \times monkey)} = 0.06$. Generalization category decoder TE: $p_{(decoding\_accuracy \times monkey)} = 0.18$, TEO: $p_{(decoding\_accuracy \times monkey)} = 0.17$). Decoding accuracy in TE was significantly more strongly associated with behavior than in TEO (one-tail 95% CI lower bound = 0.07, 0.17 for category decoder and generalization category decoder, respectively, bootstrapping Pearson correlation). Furthermore, pooling TEO with TE did not enhance behavioral prediction when regressing behavior on neural d′. Information-criterion measures, AIC (Akaike Information Criterion) and BIC (Bayesian Information Criterion), were lowest for TE compared with TEO and the combined TE + TEO model (Category decoder: AIC 94.20 (TE), 108.55 (TEO), 97.16 (TE + TEO); BIC 96.11, 110.47, 99.08. Generalization category decoder: AIC 97.01, 107.42, 99.01; BIC 98.93, 109.33, 100.93), indicating that TEO contributes little additional learning-relevant variance across days.

Third, we performed partial-correlation analyses. Across learning, TE decoding remained significantly related to behavior after controlling for TEO and subject (partial $r = 0.87$, 0.86, $p = 2.73 \times 10^{-4}$,

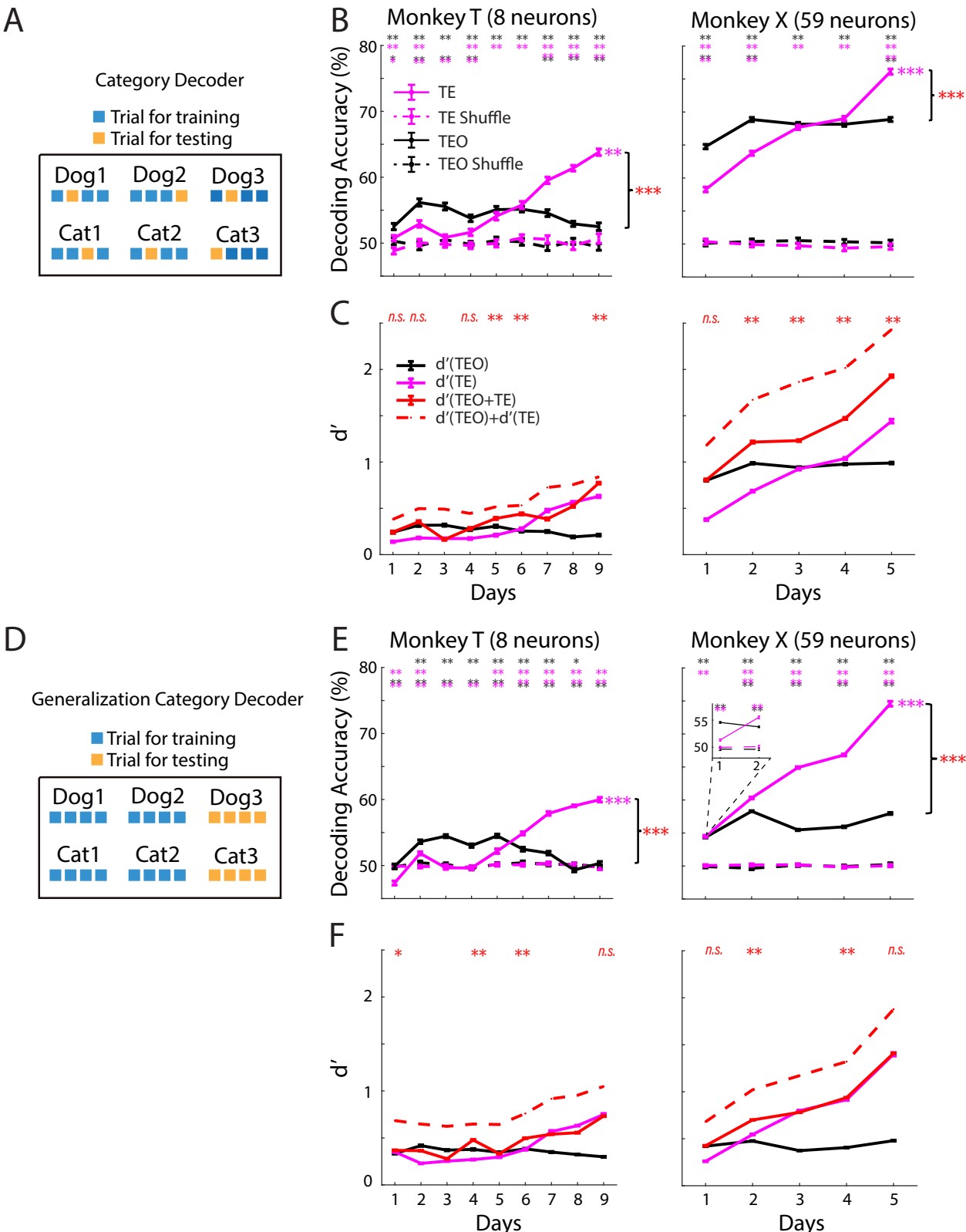

3.25 × 10⁻⁴ for category decoding and generalization category decoding, respectively). The reciprocal partial correlation for TEO (controlling for TE and subject) was not significant (partial $r = 0.2$, −0.03, $p = 0.54$, 0.92 for category decoding and generalization category decoding, respectively). Thus, TE carries unique behavior-relevant variance beyond that recorded from TEO.

Taken together, these results suggest that TE carries behaviorally relevant category information to a greater extent than does TEO.

### Choice-related signals in TE and TEO

We further quantified how well the responses of both single neurons and neuronal populations predicted the animal's category choice on a trial-by-trial basis (see Methods). We focused our choice-related analysis on 50% morph (half dog and half cat) trials, which allowed us to dissociate neural activity related to internal choices from categorical stimulus-driven responses. The proportion of neurons with significant choice encoding in either TE or TEO remained at chance level

**Fig. 3 | Category decoding accuracy rose in TE but not TEO during learning.**
**A** Illustration of category decoding. **B** Category decoding accuracy. Magenta (black) asterisks: above-chance accuracy for TE (TEO); black-over-magenta (magenta-over-black): TEO > TE (TE > TEO). T: $p_{(TEO\text{-vs.-shuffle})} = 0.003$ (day 1), 0.0011 (day 2–9); $p_{(TE\text{-vs.-shuffle})} = 0.0039, 0.0015, 0.085, 0.0067$ (day 1–4), 0.0015 (day 5–9); $p_{(TEO\text{-vs.-TE})} = 0.01, 0.0018, 0.0018, 0.003, 0.21, 0.39, 0.0018, 0.0018, 0.0018$; X: $p_{(TEO\text{-vs.-shuffle})} = 0.001$, $p_{(TE\text{-vs.-shuffle})} = 0.001$ for all days; $p_{(TEO\text{-vs.-TE})} = 0.0017, 0.0017, 0.23, 0.09, 0.0017$ (day 1–5), 1000 permutations, two-sided, same across figures. TE: 99% CI of $r_{(TE)} - r_{(shuffle)}$: [0.02 1.11] (T); 99.9% CI of $r_{(TE)} - r_{(shuffle)}$: [0.04 1.98] (X). TEO: 95% CI of $r_{(TEO)} - r_{(shuffle)}$: [−0.71 0.45]; 95% CI of $r_{(TEO)} - r_{(shuffle)}$: [−0.21 1.55], bootstrapping Pearson correlation (5000 bootstrap samples, two-sided, same across figures). $p_{(region \times day)} = 1.8 \times 10^{-76}$ (T), $3.1 \times 10^{-83}$ (X), LM. **C** Category discriminability from category decoder. Red asterisks: $d'(TEO + TE) > \max[d'(TEO), d'(TE)]$; n.s.: no difference. Other days: $d'(TEO + TE) < \max[d'(TEO), d'(TE)]$. T:

$p = 0.94, 0.059, 0.0018, 0.58, 0.0018, 0.0018, 0.0018, 0.009, 0.0018$ (day 1–9); X: $p = 0.32, 0.0018, 0.0018, 0.0018, 0.038$ (day 1–5), permutation test. **D** Illustration of generalization category decoding. **E** Generalization category decoding accuracy. T: $p_{(TEO\text{-vs.-shuffle})} = 0.88$ (day 1), 0.0015, (day 2–7), 0.027, 0.19 (day 8–9); $p_{(TE\text{-vs.-shuffle})} = 0.0013$ (day 1–2, 5–9), 0.51, 0.77 (day 3–4); $p_{(TEO\text{-vs.-TE})} = 0.001$ (all days); X: $p_{(TEO, TE \text{ vs. shuffle})} = 0.001$ (all days); $p_{(TEO\text{-vs.-TE})} = 0.4$ (day 1), 0.0012 (day 2–5), permutation test. Inset: two equal sessions, day 1. TE: 99.9% CI of $r_{(TE)} - r_{(shuffle)}$: [0.01 1.68] (T), [0.04 1.97] (X). TEO: 95% CI of $r_{(TEO)} - r_{(shuffle)}$: [−0.9 0.07], [−0.48 0.76], bootstrapping Pearson correlation. $p_{(region \times day)} = 4.9 \times 10^{-176}$ (T), $4.5 \times 10^{-240}$ (X), LM. **F** $d'$ from generalization category decoder. T: $p = 0.032, 0.0018, 0.0018, 0.0018, 0.038, 0.0018, 0.032, 0.0018, 0.12$ (day 1–9); X: $p = 0.53, 0.0025, 0.02, 0.0025, 0.55$ (day 1–5), permutation test. Data are mean ± SEM; $p$ values FDR-adjusted (Benjamin–Hochberg). *$p < 0.05$, **$p < 0.01$, ***$p < 0.001$.

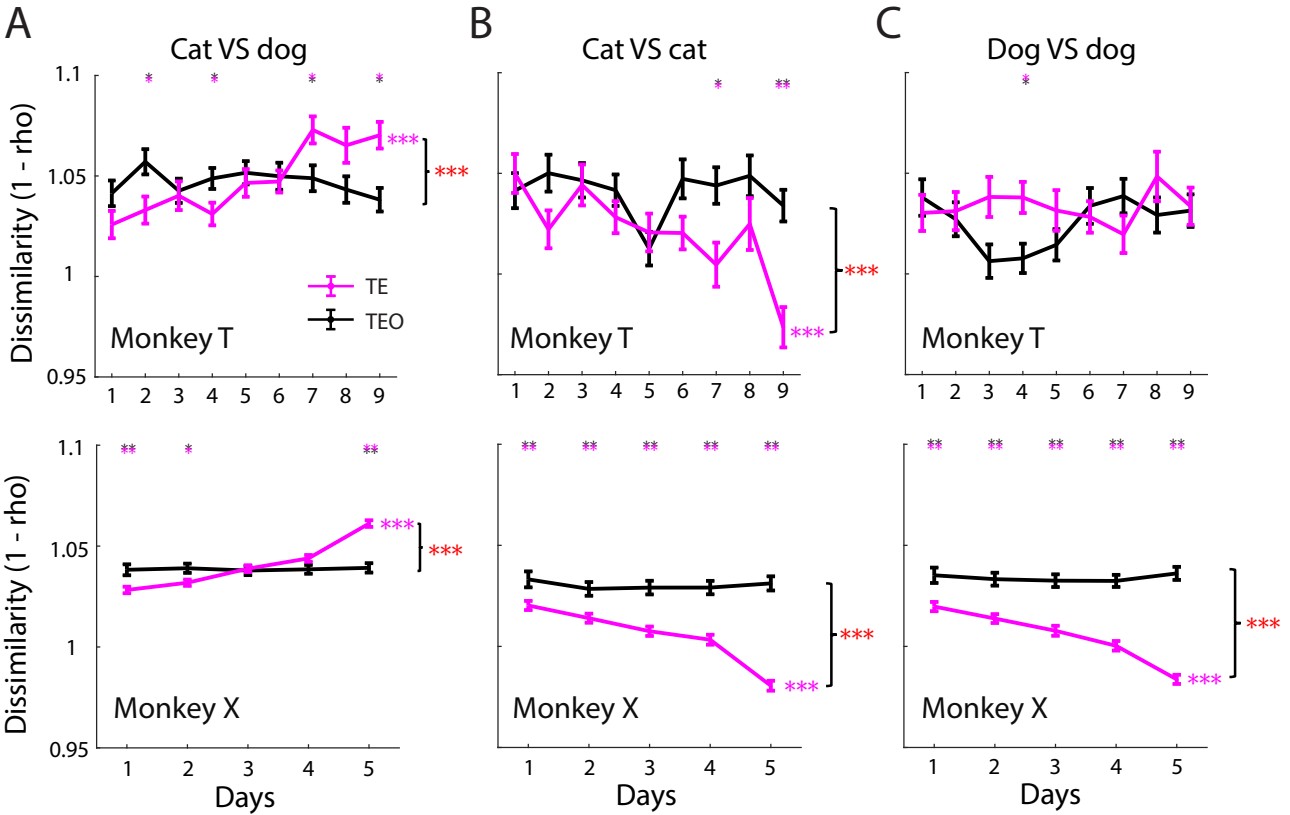

**Fig. 4 | Representational dissimilarity for image pairs between or within categories.** **A** Cat-vs.-dog: T: $p_{(dissimilarity\text{-vs.-day})} = 1.49 \times 10^{-11}$ (TE), 0.39 (TEO), $\beta_1 = 5.9 \times 10^{-3}, -6.8 \times 10^{-4}$, LM. $p_{(TEO\text{-vs.-TE})} = 0.12, 0.03, 0.78, 0.049, 0.73, 0.78, 0.027, 0.061, 0.018$ (day 1–9), permutation test; X: $p_{(dissimilarity\text{-vs.-day})} = 2.79 \times 10^{-51}$ (TE), 0.86 (TEO), $\beta_1 = 7.8 \times 10^{-3}, 1.4 \times 10^{-4}$, LM. $p_{(TEO\text{-vs.-TE})} = 0.005, 0.022, 0.75, 0.071, 0.005$ (day 1–5), permutation test. $p_{(region \times day)} = 2.5 \times 10^{-8}$ (T), $4 \times 10^{-17}$ (X), LM. **B** Cat-vs.-cat: T: $p_{(dissimilarity\text{-vs.-day})} = 6.06 \times 10^{-7}$ (TE), 0.63 (TEO), $\beta_1 = -6.4 \times 10^{-3}, -5.6 \times 10^{-4}$, LM; $p_{(TEO\text{-vs.-TE})} = 0.57, 0.08, 0.85, 0.36, 0.57, 0.081, 0.027, 0.27, 0.009$ (day 1–9), permutation test; X: $p_{(dissimilarity\text{-vs.-day})} = 9.66 \times 10^{-34}$ (TE), 0.76 (TEO),

$\beta_1 = -9 \times 10^{-3}, -3.3 \times 10^{-4}$, LM. $p_{(TEO\text{-vs.-TE})} = 0.006, 0.0025, 0.0017, 0.0017, 0.0017$ (day 1–5), permutation test. $p_{(region \times day)} = 6.6 \times 10^{-4}$ (T), $8.7 \times 10^{-11}$ (X), LM. **C** Dog-vs.-dog: T: $p_{(dissimilarity\text{-vs.-day})} = 0.81$(TE), 0.27(TEO), $\beta_1 = 3 \times 10^{-4}, 1.2 \times 10^{-3}$, LM; $p_{(TEO\text{-vs.-TE})} = 0.018$ for day 4, $p > 0.05$ for other days, permutation test; X: $p_{(dissimilarity\text{-vs.-day})} = 3.04 \times 10^{-31}, 0.92$, $\beta_1 = -8.6 \times 10^{-3}, 1 \times 10^{-4}$, LM. $p_{(TEO\text{-vs.-TE})} = 0.001$ for all days, permutation test. $p_{(region \times day)} = 0.59$ (T), $1 \times 10^{-11}$ (X), LM. Number of image pairs: T = 2250 (cat-vs.-dog), 1125 (cat-vs.-cat), 1125 (dog-vs.-dog). X = 9500, 4750, 4750. Data are mean ± SEM; $p$ values FDR-adjusted (Benjamin–Hochberg). *$p < 0.05$, **$p < 0.01$, ***$p < 0.001$.

(Fig. S5A). This result is consistent with previous findings[32], suggesting that single-neuron activity in IT may not reliably predict category choices for ambiguous stimuli.

At the population level, we observed a different pattern. Significant choice-related decoding was observed on multiple days: for monkey T, TE and TEO on 5/9 and 2/9 days, respectively; for monkey X, TE and TEO on all 5 days and 4/5 days, respectively (Fig. S5B, $p$ values were adjusted by Benjamin–Hochberg false discovery rate (FDR) correction). The proportion of days in which category choice could be

predicted from the neural responses was not statistically different between TE and TEO for either monkey (Fisher's exact test, $p > 0.05$ for both monkeys). These results suggest that, although individual neurons do not seem to encode categorical choices during ambiguous trials, population-level activity in IT cortex relatively reliably reflects the animal's decisions.

Interestingly, although category representation increased over training (Fig. 3B, E), choice signals at the population level were not shaped by learning (Fig. S5B). These results imply a functional

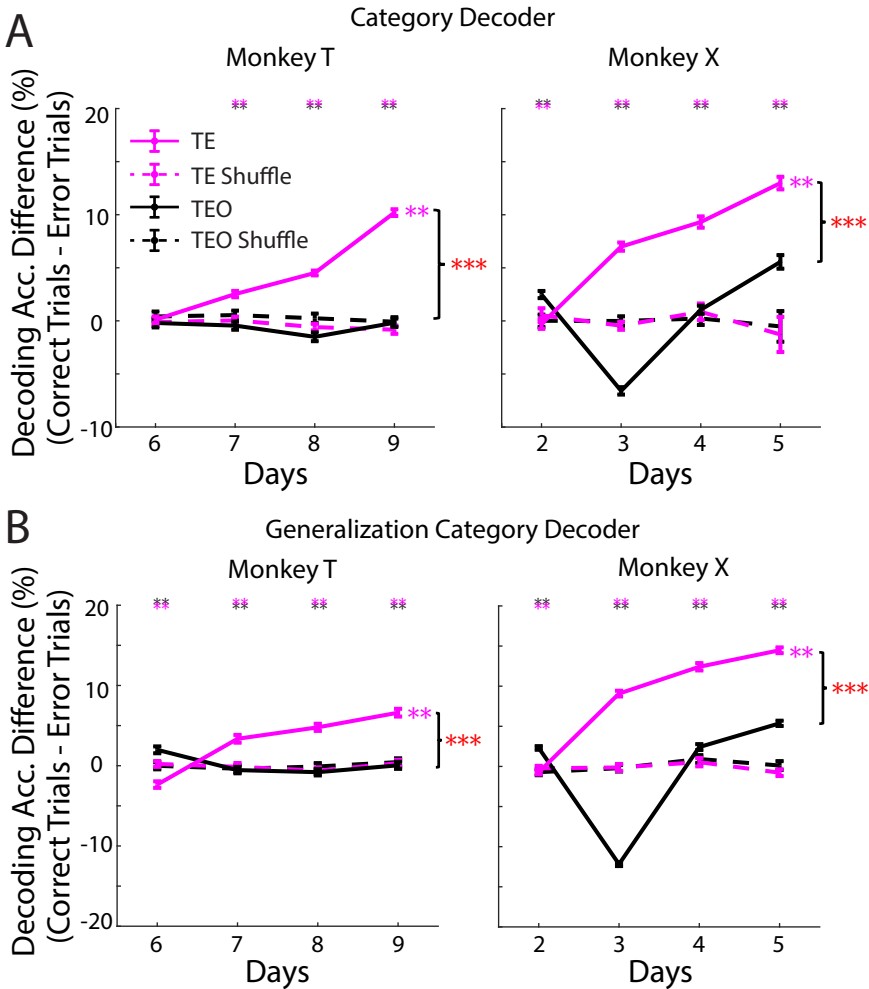

**Fig. 5 | Comparison of category decoding accuracy in correct and error trials.** **A** Difference of category decoding accuracy from category decoder between correct and error trials. Decoding accuracy difference between correct and error trials was larger in TE than TEO in most days. Black-over-magenta and magenta-over-black asterisks indicate TEO > TE and TE > TEO, respectively. T: $p_{(\text{TEO-vs.-TE})} = 0.56$, 0.0013, 0.0013, 0.0013 (day 6–9); X: $p_{(\text{TEO-vs.-TE})} = 0.001$ for all the days, permutation test. Decoding accuracy difference between correct and error trials significantly increased across learning days in TE, not TEO. TE: 99% CI of $r_{(\text{TE})} - r_{(\text{shuffle})}$: [0.27 1.97] (T), [0.01 1.95] (X); TEO: 95% CI of $r_{(\text{TEO})} - r_{(\text{shuffle})}$: [−1.05 1.24], [−0.55 1.39], bootstrapping Pearson correlation (5000 bootstrap samples, two-sided),

decoding accuracy vs. day number. $p_{(\text{region} \times \text{day})} = 1.3 \times 10^{-46}$ (T), $2.5 \times 10^{-14}$ (X), LM. **B** Similar as (**A**), but decoding accuracy was from generalization category decoder. Generalization category decoding accuracy difference between correct and error trials was larger in TE than TEO in most days. $p_{(\text{TEO-vs.-TE})} = 0.001$ for all the days in both monkeys, permutation test. Decoding accuracy difference between correct and error trials significantly increased across learning days in TE, not TEO. TE: 99% CI of $r_{(\text{TE})} - r_{(\text{shuffle})}$: [0.02 1.89] (T), [0.07 1.89] (X); TEO: 95% CI of $r_{(\text{TEO})} - r_{(\text{shuffle})}$: [−1.68 0.17], [−0.59 0.48], bootstrapping Pearson correlation. $p_{(\text{region} \times \text{day})} = 4.6 \times 10^{-43}$ (T), $6.6 \times 10^{-17}$ (X), LM. Data are mean ± SEM; $p$ values FDR-adjusted (Benjamini–Hochberg). $*p < 0.05$, $**p < 0.01$, $***p < 0.001$.

dissociation between category representation and choice-related activity in IT cortex during learning.

**TE showed stronger correct-vs.-error categorical enhancement than TEO**

The analyses described above included both correct trials and error trials. We further studied the relation of TE and TEO activity to behavioral performance by investigating the representation of category in correct trials vs. error trials. In candidate brain regions that might subserve category judgments, representation of the category should be stronger when monkeys categorize correctly than when they respond incorrectly. We focused on the days when monkeys were relatively well-trained (correct rate >75%) for two reasons: first, we could obtain enough correct trials from each image to train a decoder (see "Methods"). Second, in the early days of learning, monkeys were exploring the underlying categorization rules or boundaries, resulting in high variability in the behavioral responses. Correct trials for images

with low ambiguity (morph level: 0%, 25%, 75%, 100%) were used for decoder training. Correct and error trials from ambiguous images (morph level: 35%, 40%, 45%, 55%, 60%, 65%) were used for testing (see "Methods"). Decoding accuracy in correct trials was higher than for error trials in the TE population (Fig. S6A). Similar results were seen in a recent study[26], in which enhanced category information in TE population was found during correct, but not error trials when monkeys categorized ambiguous body and object images. For TEO, the results were not consistent between the two monkeys (Fig. S6B). For both monkeys, the decoding accuracy difference between correct and error trials was larger in TE than in TEO (Fig. 5A), which indicated greater enhancement of category representation in correct trials when compared with error trials for TE than TEO. The decoding accuracy difference increased significantly across days in TE, not TEO (Fig. 5A). The same pattern of results was obtained using a generalization decoding strategy (Figs. 5B and S6C, D). The enhanced category information in the TE population in correct trials vs. error trials indicates that TE is a

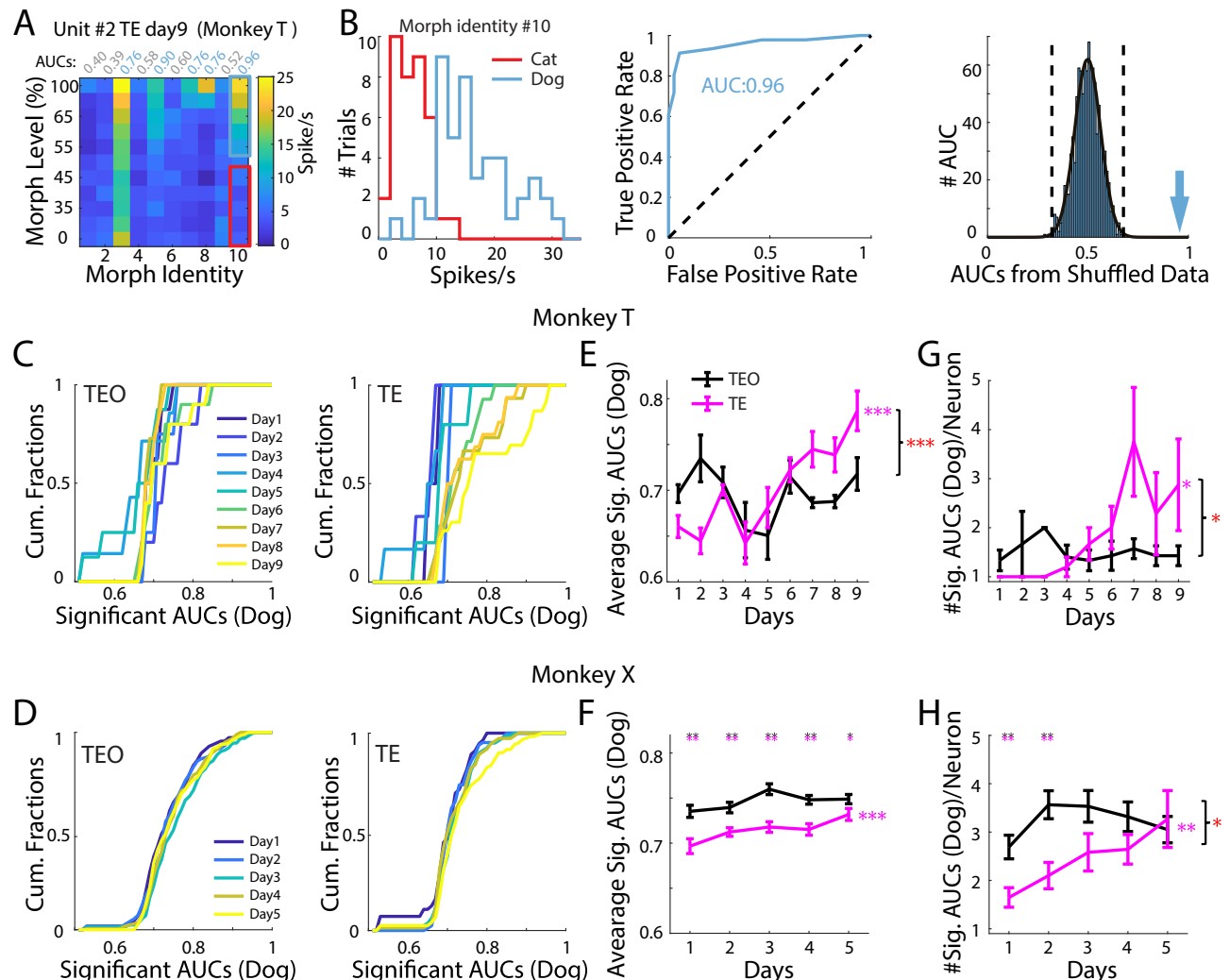

**Fig. 6 | Encoding of dog images was enhanced in TE, not TEO. A** An example neuron preferring dogs. Blue and gray indicated significant (dog) and non-significant AUCs (area under the receiver operating characteristic [ROC] curves), respectively. **B** Left, distribution of responses to cat and dog images in morph identity #10. Middle, ROC analysis on responses in the left panel. Right, AUCs from 1000 permutations. Black curve was the fitted Gaussian function. Two vertical dashed lines indicated [μ − 1.96σ, μ + 1.96σ]. **C, D** Cumulative distribution of significant dog AUCs for T and X, respectively. **E, F** Average significant dog AUCs across days in T and X. respectively. T: $p_{(AUC\text{-vs.-day})}$ = 0.81 (TEO), 3.1 × 10⁻⁶ (TE), $\beta_1$ = 0.0006, 0.02, LM (two-sided, same across panels). n(TEO) = 8, 5, 4, 7, 8, 10, 11, 10, 10 significant dog AUCs (day 1–9); n(TE) = 2, 3, 2, 6, 5, 14, 15,16, 23; $p_{(TEO\text{-vs.-TE})}$ > 0.05 for all days; X: $p_{(AUC\text{-vs.-day})}$ = 0.093 (TEO), 4.3 × 10⁻⁴ (TE),

$\beta_1$ = 0.0031, 0.0073, LM. n(TEO) = 113, 164, 166, 199, 183 significant dog AUCs (day 1–5); n(TE) = 56, 84, 80, 74, 121. $p_{(TEO\text{-vs.-TE})}$ = 0.0017, 0.0062, 0.0017, 0.0017, 0.033 (day 1–5), 1000 permutations, two-sided, same across panels. $p_{(region \times day)}$ = 9 × 10⁻⁶ (T), 0.16 (X), LM. **G, H** Number of significant dog AUCs in neurons with ≥1 significant dog AUC. T: $p_{(\#AUC\text{-vs.-day})}$ = 0.93 (TEO), 0.013 (TE), $\beta_1$ = −0.003, 0.28, LM. n(TEO) = 6, 3, 2, 5, 6, 7, 7, 7, 6 neurons (day 1–9), n(TE) = 2, 3, 2, 5, 3, 7, 4, 7, 8. $p_{(TEO\text{-vs.-TE})}$ > 0.05 for all days; X: $p_{(\#AUC\text{-vs.-day})}$ = 0.78 (TEO), 0.0015 (TE), $\beta_1$ = 0.026, 0.38, LM. n(TEO) = 42, 46, 47, 60, 60 neurons (day 1–5); n(TE) = 34, 40, 31, 28, 37. $p_{(TEO\text{-vs.-TE})}$ = 0.005, 0.005, 0.14, 0.27, 0.74 (day 1–5), permutation test. $p_{(region \times day)}$ = 0.011 (T), 0.018 (X), LM. Data are mean ± SEM; p values FDR-adjusted (Benjamini–Hochberg). *$p < 0.05$, **$p < 0.01$, ***$p < 0.001$.

---

plausible candidate region for supporting downstream computation of category membership of the stimuli used in this study. Further investigation is needed to clarify the contributions of TEO given individual variability across animals.

**Neuronal encoding of dog-like images was enhanced in TE, not TEO**
To better understand the coding properties of individual neurons in TE and TEO, the category preference was evaluated via ROC (Receiver operating characteristic) analysis performed on neural responses to each morph identity separately for each neuron. AUC (area under the ROC curves) was used to measure the separability of responses to "cat-like" and "dog-like" images. A permutation test was performed to evaluate the significance of an AUC (see examples in Figs. 6A, B and S7).

Neurons that responded preferentially to one category across all morph identities were called "category-selective" neurons, and neurons that preferred different categories depending on morph identity were called "mixed-selectivity" neurons. There were more "category-selective" neurons than "mixed-selectivity" neurons in TE (Fig. S8A, E). In TEO, results were not consistent between the two monkeys (Fig. S8B, E). The proportion of "category-selective" neurons was similar in TE and TEO, however, the proportion of "mixed-selectivity" neurons was smaller in TE than in TEO (Fig. S8E). These results indicate that TE neurons were more likely to show a preference for one of the two categories compared to TEO neurons. These results suggest that relative to TEO, neurons in TE were more likely to exhibit responses dominated by a single category. Although both areas encode category information, the reduced mixed selectivity in TE may point to more specialized category representation compared to the more multiplexed coding in TEO.

We next tested whether dog-like images became easier to differentiate from cat-like images in the same morph identity during learning. Significant AUCs from all neurons were included in this analysis. For both monkeys, encoding of dog-like images by TE neurons increased during learning, but encoding by TEO neurons did not (Fig. 6C–F. The brain region × day interaction was significant in monkey T, but not in monkey X). In contrast, encoding of cat-like images did not increase during learning in either TE or TEO (Fig. S9A–D). For the neurons preferring at least one identity of dog-like images, i.e. with at least one significant AUC for dog-like images, the number of preferred dog-like images per neuron increased during learning in TE, not in TEO (Fig. 6G, H), For the neurons preferring at least one identity of cat-like images, the number of preferred cat-like images per neuron did not show significant changes during learning in either TE or TEO (Fig. S9E, F).

Thus, as learning progressed, the neural representations of dog-like images became increasingly differentiable from the representations of cat-like images in the same morph identity in TE, while remaining unchanged in TEO. This TE-specific growth was significant in monkey T and showed a similar but non-significant trend in monkey X, indicating that the enhancement of dog-image encoding was robust in one animal and partial in the other.

### Learning-related plasticity in TE emerges only for feature-ambiguous categorization

To test whether the results described above generalized to other visual categories, we conducted additional categorization tasks with monkey X (see Supplementary Notes), including car/truck categorization using 20 cars and 20 trucks (2 days), and 240 cars and 240 trucks (1 day), as well as cat/dog categorization without morphing (20 cats and 20 dogs for 3 days; 240 cats and 240 dogs for 1 day). The task design was the same as Fig. 1A for monkey X. Cat or car trials were not rewarded, dog or truck trials were rewarded for a release on green.

We replicated the main findings using the non-morphed 20-cat/20-dog categorization tasks (Fig. S10). Decoding accuracy in area TE, but not TEO, increased across learning sessions (Fig. S10D). Critically, decoding accuracy of category decoder in TE—but not in TEO—was positively linked to behavioral performance (bootstrapped Pearson correlation, 95 % CI lower bounds: 0.01 for TE, −0.55 for TEO; brain region × decoding accuracy interaction: $p = 1.2 \times 10^{-14}$, LM. For the generalization category decoder, TE accuracy showed a positive yet non-significant trend: 95% CI lower bound = −0.06, whereas TEO again exhibited no association: lower bound = −0.48; interaction: $p = 1.2 \times 10^{-38}$). TE and TEO carried complementary and partially independent category information on multiple days (Fig. S10E, 4/6 days for both decoders). TE, but not TEO, population response patterns evoked by dog images became more similar over training, while cat-related patterns did not show such improvement in either TE or TEO (Fig. S10F). This asymmetry may reflect the fact that this was the monkey's first exposure to a cat/dog categorization task. Given the task's asymmetric reward structure and the limited training period (3 days), there may not have been sufficient time for representational changes for cat images. Moreover, categorical representations in TE—but not TEO—were enhanced during correct compared to error trials (Fig. S10G, H). At the single-neuron level, TE responses became more evenly distributed[33] across both cat and dog images with learning. No evidence of learning-related changes was observed in TEO (Fig. S10I).

However, for the 20-car/20-truck categorization task, we failed to detect any systematic changes in either TE or TEO responses across learning (Fig. S11). This likely reflects the lower visual ambiguity of the car/truck stimuli compared to the cat/dog images. Cars and trucks differ along relatively coarse, low-level visual features such as shape and geometric contours, which may be sufficiently processed by earlier visual areas. In contrast, categorizing cats and dogs—especially when morphed—requires fine-grained discrimination of subtle, overlapping features, which require greater demands on high-level visual processing. We also observed TE and TEO carried complementary, partially independent category information on all days (Fig. S11E).

We challenged the monkey with a large set of novel images presented in a single day (240-cat/240-dog, 240-car/240-truck). The results were consistent with the above findings. A small but significant increase in decoding accuracy across repeats was observed in TE, but not in TEO, for the cat/dog categorization task (Fig. S12C). In contrast, neither TE nor TEO showed any significant increase in decoding accuracy across repeats for the car/truck categorization task (Fig. S12G). TE and TEO also carried complementary, partially independent category information in multiple repeats across both tasks (2/5 for cat/dog; 3/5 for car/truck; Fig. S12D, H), suggesting complementary contributions to category representation during these repeats. These findings suggest that TE is selectively engaged, and exhibits plasticity, only when the visual categorization demands fine discrimination between perceptually similar and ambiguous stimuli, such as cat and dog images.

## Discussion

The ability of humans and monkeys to form categories of visual images is thought to rely on neural plasticity in the ventral visual pathway. No prior studies have tracked the effect of neural plasticity on IT neural responses during categorization learning in nonhuman primates, presumably due to the logistical difficulty of doing so over several months of learning. In the present study, we reduced the learning phase of a visual categorization task to a few days by introducing a pre-training procedure which made it possible to track the neural plasticity during category learning. The implanted high-channel-count microelectrode arrays allowed simultaneous recording from two subregions at the end of ventral visual pathway, area TE and area TEO. We found that neural plasticity in TE, but not TEO, reflected monkeys' behavior in learning a visual categorization task.

Mechanistically, we interpret our results as evidence that plasticity in TE supports the formation of more abstract, experience-dependent, behaviorally related category representations. In contrast, TEO may support simpler visual feature processing, which is less plastic, and is hence less influenced by learning.

Based on our findings—that category-related signals in TEO are stronger than those in TE during the early stage of learning (Figs. 3B, E and 4A), that TE alone shows a gradual, learning-related increase in decoding accuracy, we propose that TE and TEO play complementary roles in category learning. Representation of category in both TE and TEO is significantly stronger than chance level (Fig. 3B, E). In both monkeys, and across decoding strategies, the category discriminability of pooled TE and TEO neurons—i.e., $d'(TE + TEO)$—often exceeded that of the stronger individual region (Figs. 3C, F and S10E, S11E, S12D, H). Specifically, at the initial stages of learning, TEO may provide an early, coarse category representation sufficient for basic categorization demands, perhaps based on simple visual features. As learning progresses, TEO, as well as V4 which also directly projects to TE[14], continues to supply lower-level visual feature representations, which are refined and integrated within TE to gradually construct more abstract and behaviorally relevant category representations. This interpretation is consistent with prior lesion studies showing that bilateral removal of either TE or TEO causes only mild deficits, while combined lesions produce severe, long-lasting impairments, highlighting the interdependent roles of these two regions.

It is important to consider the potential impact of extensive pre-training on the initially stronger category representation observed in TEO during the first day of the morphed cat/dog categorization task. Prior to the morphed cat/dog categorization task, monkeys underwent months of training to become familiar with the task structure, fixation requirements, and general categorization demands. They were trained

on non-morphed cat/dog categorization tasks before progressing to the morphed versions. For monkey X, we recorded data during the monkey's first exposure to cat/dog categorization task (20-cat/20-dog, Fig. S10). Even in this early stage, decoding accuracy was consistently higher in TEO than in TE, suggesting that the strong initial category representation in TEO is not solely attributable to extensive pre-training on cat/dog tasks. However, we cannot entirely rule out the influence of other pre-training experiences, such as fixation training or earlier exposure to Walsh pattern categorization tasks, which may have contributed to the development of initial category-selective responses in TEO.

The greater plasticity of TE compared to TEO during learning may stem from its anatomical position and broader connectivity. TE neurons have larger receptive fields and integrate bottom-up inputs from simple visual features, allowing them to respond to complex stimuli and support more abstract, flexible representations[14,34]. In contrast, TEO neurons have smaller receptive fields and respond to simpler features, providing stable encoding of low- to mid-level visual features[14,35]. Moreover, TE exhibits broader connectivity than TEO, including reciprocal connections with the medial temporal lobe, orbitofrontal cortex, and ventrolateral prefrontal cortex—regions involved in memory, reward processing, and decision-making[14]. These widespread connections may facilitate top-down modulation in response to behavioral relevance, enabling TE to adapt its representations through learning. In contrast, the more limited connectivity of TEO may constrain its capacity for such learning-dependent plasticity.

The observed enhanced category decoding accuracy in TE, along with an increase in TE population similarity for stimuli within the same category, suggest that learning sculpts TE representations to better reflect categorical structure. One possible mechanism underlying this reorganization is enhanced encoding of shared visual features among category members. This aligns with prior findings showing that attribute tuning is shaped by learning[27,28] and that population-level category decoding can arise even from neurons primarily tuned to simple shape features, without requiring explicit categorical coding[36]. Thus, while such reorganization supports category-based discrimination, it does not necessarily imply that TE computes category membership per se. Instead, TE may provide structured sensory representations that facilitate downstream decision-making processes.

Although category decoding accuracy in TE increased significantly over learning and correlated with behavioral performance, trial-by-trial choice predictability remained modest and unchanged. This dissociation suggests that learning primarily enhances the representation of category structure in TE, while the final decision continues to be shaped by additional variables introduced downstream. Thus, TE provides information sufficient to support improved choices but is not the exclusive causal driver of behavioral output. Downstream areas may integrate the category information from TE with other signals (e.g., attention, priors, motor noise) to generate the final decision.

De Baene et al.[30] found that category information significantly increased after visual category learning in TE. However, the category information before learning was measured in a passive viewing task, and the category information after learning was from an active categorization task, thus the differences observed may have been confounded by task-dependent effects[37]. We recently compared the category information in TE and TEO before and after visual category learning with a passive viewing task and found that category learning increased the category information in TE[31]. However, due to the asymmetric reward structure of our previous task, we could not rule out a role for reward associations in these changes. In the current study, we balanced the value of reward associated with the two categories by time discounting in one monkey (monkey T) and found similar results to the monkey performing the categorization task with

an asymmetric reward structure. Thus, we conclude that visual category learning can increase category information in TE, not TEO. Although we emphasize the key function of TE in categorization learning here, our lesion studies have shown that monkeys with TE removal can still slowly improve their performance in categorization tasks with training[10–12]; in the absence of TE, TEO may have made a compensatory contribution to the improvement.

Due to the unbalanced 2-IFC task design, the category information presented in the current study may be affected by different contingencies, e.g., contaminated by motor preparation, reward expectation, motivation, or arousal state. Neural activity in TE is unlikely to be modulated by motor preparation because previous studies found little motor dependent information in TE[32]. However, neural activity in TE does represent the expectation of reward outcomes[32,38]. Further, different anticipation of reward outcomes may cause different levels of motivation and arousal, which could modulate the neural responses in TE. The confound of unbalanced reward may exist for monkey X, but is unlikely to be relevant to the interpretation of data collected from monkey T. For monkey T, the subjective value of the outcome associated with each of two categories was balanced, indicated by unbiased choices when the images were from morph level 50% (Fig. S1C). With this balanced reward contingency in monkey T, we found similar results to those observed in monkey X.

Encoding of dog-like images, not cat-like images, increased during learning in both monkeys, presumably, in part, due to the asymmetry of contingencies in the 2-IFC task. However, as discussed above, it is unlikely that increased encoding strength was caused solely by unbalanced outcomes, motivation, or arousal. There are at least three alternative explanations for the increased encoding of dog-like images: first, TE of nonhuman primates may inherently prefer processing dog images to cat images. This is unlikely because we did not observe consistently more TE neurons encoding dog-like images than cat-like images on the first day of testing (Fig. S8C). On the contrary, we previously found more TE neurons responded to cats than dogs[31]. Further, the encoding strength of dog-like images was not stronger than cat-like images on the first day (Fig. 6E, F vs. Fig. S9C, D. For monkey T TE, encoding strength of cat-like images was significantly stronger than that of dog-like images. $p = 0.023$, $t_{(5)} = -3.24$; For monkey X TE, $p = 0.99$, $t_{(94)} = 0.009$). Second, the unbalanced reward structure could cause the monkeys to attend more to dog-like images. In monkey X, dog-like images were always associated with rewards and cat-like images were not associated with rewards. The reward availability could have biased monkey X's attention to dog-like images. In monkey T, although the subjective value of two outcomes was similar, the amount of fluid associated with dog-like images was larger than cat-like images, which could have caused monkey T to attend preferentially to dog-like images. Third, monkeys could solve the cat-dog categorization task with a "dog"-or-"not dog" strategy. It is not necessary for monkeys to recognize or memorize images in both categories. One effort-saving strategy would be to only recognize or memorize images in one category. Then monkeys could complete the categorization using a "dog"-or-"not dog" or "cat"-or-"not cat" strategy. Coincidental selection of the same "dog"-or-"not dog" strategy by two monkeys could also bias encoding to dog-like images in both monkeys.

Different training procedures may have encouraged the two monkeys to adopt different categorization strategies: an A/not-A strategy (only one category rewarded) for monkey X, and an A/B strategy (both categories rewarded) for monkey T. While we cannot directly confirm the strategies monkeys used, the reward structure likely biased the animals toward these approaches. In human studies, A/not-A and A/B categorization tasks are known to recruit distinct learning systems: A/not-A tasks tend to rely more on the striatum and perceptual learning systems, whereas A/B tasks more strongly engage the hippocampus and declarative memory systems[39,40]. However, our findings reveal consistent TE-related plasticity during learning across

two monkeys, suggesting that in non-human primates, visual categorization involving ambiguous stimuli and fine-grained discrimination may ultimately recruit overlapping neural processes in the IT cortex—regardless of the learning strategy. This possibility warrants further investigation in future studies.

We observed several inter-animal differences including: decision biases in the initial days (Fig. S1C), proportion of neurons modulated by category (Fig. 2E), neurons modulated by morph level/identity (Fig. S3), category selectivity neurons (Fig. S8B, E), dog generalization (Fig. 4C), dog-image encoding (Fig. 6E, F) and error effects for TEO (Fig. S6B, D). These differences may stem from differences in task design (e.g., reward contingencies), species differences, and/or the unequal number of recorded neurons—substantially fewer neurons were recorded from monkey T, which limits our statistical power to detect learning-related changes. Despite these individual differences, our key finding—enhanced neural plasticity in TE compared to TEO during learning of our ambiguous visual categorization task—was consistent in most analyses across both subjects, demonstrating that the core effects generalize across animals despite individual variability.

Both monkeys increased psychometric slope-indicating a shared gain in category sensitivity-but their learning diverged in how the decision boundary evolved: Monkey T's slope rose steadily with little boundary shift, whereas Monkey X coupled its slope increase to an early, pronounced boundary realignment. Despite this behavioral divergence, both animals exhibited parallel gains in category information within area TE, suggesting that TE plasticity may provide a common representational substrate that can be leveraged by either strategy-rule discovery or sensory sharpening-to support successful visual categorization.

In summary, our results reveal TE as the principal locus of experience-dependent plasticity that converts ambiguous visual features into behaviorally meaningful categories, whereas TEO provides a stable, relatively low-level feature-based representation. These findings position TE as a key node in the transformation of sensory representations into categorical knowledge, providing important insights into the neural mechanisms by which experience shapes perceptual decision-making.

## Methods

### Subjects

We collected data from one adult male rhesus macaque (*M. mulatta*, monkey T, 6-year-old, 10 kg) and one adult male Japanese macaque (*M. fuscata*, monkey X, 13-year-old, 11 kg, the same individual referred to as monkey X in our previous study[31]). For monkey T, all experimental protocols conformed to the Institute of Medicine Guide for the Care and Use of Laboratory Animals and were performed under an Animal Study Proposal approved by the Animal Care and Use Committee of the National Institute of Mental Health. For monkey X, all surgical and experimental procedures were approved by the Animal Care and Use Committee of the National Institute of Advanced Industrial Science and Technology (Japan) and were implemented in accordance with the "Guidelines for Proper Conduct of Animal Experiments" (Science Council of Japan) and "Guidelines for the Care and Use of Nonhuman Primates in Neuroscience Research" (2021 edition, Japan Neuroscience Society).

### Behavioral training

The monkey sat in a primate chair with its head restrained, facing a LCD monitor[41] in a darkened, sound-attenuated chamber. The distance between the monitor and monkey's eyes was 57 cm. The behavioral task was controlled by the real-time experimental system "REX"[42]. Visual stimuli were presented using commercially available software (Presentation, Neurobehavioral Systems).

Monkeys were initially trained to press and release a touch sensitive bar to obtain fluid reward. After this initial shaping, a red/green

discrimination task was introduced. To initiate a trial, the monkey needed to press the bar. After 100 ms, a small red square (0.5° × 0.5°) was presented in the center of the screen. To get reward, the monkey needed to continue to press the bar until the red square turned to green, which happened randomly within 500–1500 ms after the red square appeared. The response window was 150–1000 ms after the green square appeared. Releasing the bar in the response window was followed by visual feedback (a blue square replaced the green one). Reward delivery occurred 200–400 ms after the visual feedback. There was a 1 s inter-trial interval regardless of the outcome of the previous trial.

After monkey T reached criterion in the red/green task (correct rate ≥85% in 2 consecutive days), a visual discrimination task with two black and white "Walsh" patterns was introduced. A white square was presented 50 ms after the monkey touched a bar. The monkey was required to hold fixation in 0.5° × 0.5° window for 50 ms. A visual pattern (10° × 10°) was presented in the center of screen[11]. A red square appeared 350–400 ms after the visual pattern and changed to green after 2000–4000 ms. One of the patterns indicated that the bar should be released while the red square was present; doing so led to a small fluid reward. The other pattern indicated that the bar should be released while the green square was presented; doing so led to a large fluid reward. Bar-release errors led to a 4 s time-out. Once the correct rate reached 75% for 2 consecutive days, the monkey progressed to the categorization training tasks. The first training task required the categorization of 20 cat and 20 dog images. The trial structure was the same as the "Walsh" pattern discrimination task described above. To obtain reward, monkeys released the bar during the presentation of the red square when a cat image was presented, or during the presentation of the green square when a dog image was presented. After monkeys reached 75% correct in a single day, they were moved on to the second categorization training task with a larger set of, initially trial-unique, images (240 cats and 240 dogs). Morphed cat/dog images were introduced after the monkeys reached 75% correct with the larger cat/dog image set. To balance the subjective value of the outcomes, the reward size was adjusted online according to monkey T's performance history beginning from the Walsh pattern discrimination task. Each session began with 3 drops of reward available for a release on red for cat-like images and 4 drops for a release on green for dog-like images. The drop size for either choice would increase by 1 drop if the performance on this choice was lower than that on the other choice and decrease by 1 drop in the opposite condition. The lower and upper limit of drop size was 1 drop and 10 drops. The performance was calculated and updated by every 20 trials.

For monkey X, a variation of the same task design was used. If the bar was released during the presentation of the red square (first interval), there was no visual feedback, reward delivery or time-out. The monkey skipped the next trial. The optimal response strategy for monkey X depended on the visual stimulus, which was the same as for monkey T. For cat-like images (which were never rewarded), releasing early allowed the monkey to efficiently skip the non-rewarding trial. In contrast, for dog-like images (which could lead to reward), it was advantageous to wait until the green square (second interval) before releasing the bar to obtain a reward.

The use of different procedures across monkeys was not an intentional manipulation aimed at testing the distinction between A/notA and A/B, but rather a result of the progressive development of our training protocols. Monkey X was trained earlier using a more traditional A/notA design. For monkey T, we refined the procedure based on our prior experience and implemented a more balanced A/B design to reduce potential category biases.

### Visual cues

The cat/dog images in this study were the same as used in our previous reports[11,12], which were PCX format photos with 200 × 200 pixels.

Twenty (monkey T) or 10 (monkey X) pairs of cats and dogs were used to create cat-dog morph identities using Fantamorph software (Abrosoft, Beijing, China). For each morph identity, stimuli consisting of the following ratios of dog:cat were created: 100:0, 75:25, 65:35, 60:40, 55:45, 50:50, 45:55, 40:60, 35:65, 25:75, and 0:100. The distribution of the ratios was concentrated around the category boundary (dog 50:50 cat).

## Surgeries

Surgeries were performed under aseptic conditions in a fully equipped operating suite with veterinary supervision. Animals were sedated by ketamine (10 mg/kg, i.m.) before surgery. During surgery, anesthesia was maintained with isoflurane (1–4%, to effect). Body temperature, heart rate, blood pressure, $SpO_2$ and expired $CO_2$ were monitored throughout all the procedures. Surgical procedures were similar to those described previously[43]. For both monkeys, a bone flap was temporarily removed and the dura reflected to expose the left temporal cortex. For monkey X, 3 microelectrode arrays (Utah arrays, iridium oxide, 96 electrodes, 10 × 10 layout, 400 μm pitch, 1.5 mm depth, Blackrock Microsystems, Salt Lake City, USA) were implanted in the anterior, middle, and posterior parts of area TE and one in TEO in the left hemisphere. For monkey T, two Utah arrays were implanted in the anterior (64 electrodes) and posterior (96 electrodes) parts of area TE and one in TEO in the left hemisphere. Connectors for the arrays were attached to the anterior part of the skull and protected by a chamber.

## Electrophysiology recording

In each session, neural signals from Utah arrays in area TE and TEO were amplified, digitized, and filtered using the Grapevine System (Ripple, Salt Lake City, UT) with 40 kHz sample rate. Action potentials were detected in real-time and stored, which were then sorted offline via Plexon Offline Sorter. For the putative single units used in current study, firing rates exceeded 0.5 Hz, and the percentage of inter-spike intervals shorter than 2 ms was less than 1%.

## ANOVA and effect size

Response rate for each stimulus was evaluated based on the initial presentation window (monkey T: 0–550 ms; monkey X: 0–530 ms), which was determined by the earliest bar-releasing time in the first interval (Fig. S1E). Individual images were presented 6–14 times for monkey T and 9–10 times for monkey X in 1 day.

To evaluate whether a neuron was significantly modulated by image category, we performed a nested ANOVA on the response rate for each neuron. Category, morph identity, morph levels (nested in categories), and interaction between two of them were used as factors in the nested ANOVA.

To quantify how strongly neural responses were modulated by category, we computed the omega-squared ($w^2$) effect size based on outputs from a one-way ANOVA. Omega-squared is an unbiased estimator of the proportion of variance in neural activity that is attributable to a specific factor (e.g., visual category), while accounting for degrees of freedom and error variance. It is calculated as:

$$w^2 = \frac{SSQ_{effect} - df_{effect}MS_{error}}{SSQ_{total} + MS_{error}} \quad (1)$$

$SSQ_{effect}$ represents the sum of squares of the factor of interest (e. g. category), $df_{effect}$ is the degrees of freedom associated with that factor, $MS_{error}$ is the mean squared error and $SSQ_{total}$ is the sum of squares across all factors. Omega-squared values range from −1 to 1, with higher positive values indicating stronger modulation of neural responses.

## Linear model

To test whether a variable (i.e., effect size of visual category, AUC, number of significant AUC per neuron, representational dissimilarity) significantly increased across learning days, a linear regression model was fitted (*fitlm* function in MATLAB):

$$Metric_i = \beta_0 + \beta_1 * day_i + \varepsilon \quad (2)$$

day was coded as 1, 2, 3 … Significant and positive coefficient $\beta_1$ indicated a significant increase across learning days.

To test whether category learning progressed differently in TE vs. TEO, we fitted linear regression models that included a brain region × day interaction term:

$$Metric_{ij} = \beta_0 + \beta_1 * region_i + \beta_2 * day_j + \beta_3 * \left(region_i \times day_j\right) + \varepsilon \quad (3)$$

where region = 0 for TEO and 1 for TE. Significant region × day interactions indicate divergent learning patterns in TE vs. TEO, and these are marked in the figures by red asterisks positioned next to the brace that spans the two curves.

For analysis without repeated measurements (e.g. proportion of significant neurons across days, Fig. 2E), we compared learning patterns in TE and TEO by bootstrap-resampling the observations and computing the distribution of the difference between their Pearson correlation coefficients.

## ROC analysis

We used ROC analysis to quantify how well the activity of a single neuron differentiated dog-like images (or cat-like images) from images in the other half of the morph identity. For each neuron, we first extracted firing rates during a predefined post-stimulus time window across trials for images within each morph identity. Trials were grouped according to the stimulus category, defined based on the morph level (Fig. 1B). Trials with 50% morph level were not included.

We then generated an ROC curve by varying a threshold over the neuron's firing rate distribution and calculating the True Positive Rate (TPR) and False Positive Rate (FPR) at each threshold. Here, TPR reflects the proportion of "dog-like" trials in which the neuron's firing exceeded the threshold, while FPR reflects the proportion of "cat-like" trials in which firing also exceeded that same threshold. The AUC was calculated as the integral of the ROC curve and serves as a measure of how well the neuron discriminated between the two stimulus categories based on firing rate alone. AUC of each morph identity for each neuron was calculated using the *perfcurve* function in MATLAB, with dog-like images as the positive class. To evaluate whether an AUC was significant, permutation test was performed (Fig. 6B). We shuffled the category IDs for 1000 times and calculated AUC each time. We then fitted the distribution of the AUCs from shuffled data with a Gaussian function. If the actual AUC located outside [μ − 1.96σ, μ + 1.96σ] of this Gaussian distribution, it was significant. Otherwise, it is not significant. An AUC more than μ + 1.96σ indicated a significant dog-preferred neural response while an AUC less than μ − 1.96σ indicated a significant cat-preferred neural response. To make AUCs comparable between dog-like images and cat-like images, we used 1 − AUC to quantify the significant encoding of cat-like images.

## Representational dissimilarity

To compare the population response patterns elicited by two images, we calculated representational dissimilarity[25]. The dissimilarity between two response patterns was measured by correlation distance, 1 − r (Pearson correlation). The correlation was calculated across neural populations in TE or TEO. Responses of each neuron were z-scored before Pearson correlation was computed. Pairs of images were taken from different morph identities to avoid the confounds of visual similarity within the same morph identity. Representational

dissimilarity of all the possible pairs of images within one category or between two categories were computed and averaged.

## Category decoding in the neural population

The population decoding analysis was performed with a one-layer network containing one neuron (created by the *patternnet* function in MATLAB) on the simultaneously recorded neural population of TEO or TE each day. The training function was *scaled conjugate gradient* and the function used to measure the network's performance was *cross-entropy*. For each neuron, responses were *z*-scored before being fed to the network.

For the traditional decoding strategy (Fig. 3A), 6 randomly selected trials (monkey T) or 9 trials (monkey X) of each image (not including 50% morphed images) were used. These trials were split into 3 sets: training set (T: 4 trials; X: 7 trials), validation set (1 trial for both monkeys) and testing set (1 trial for both monkeys). To prevent over-fitting, the validation set was used to validate the network during training. Training was stopped when (1) validation error increased for successive 6 iterations (the weights and biases at minimum validation error were returned), or (2) gradient was less than $1 \times 10^{-7}$, or (3) number of epochs reached 10,000. To directly compare the decoding accuracy across days and brain areas, we randomly selected same number of neurons from each day and each brain area, which was determined by the minimal neuron number recorded across days and areas (monkey T: 8 neurons; monkey X: 59 neurons). For the shuffled decoding, category labels in training sets were randomly shuffled while category labels in validation sets and test sets were unchanged. The decoding procedure above was repeated 100 times for each learning day.

For the generalization decoding strategy (Fig. 3D), we used 10-fold cross-validation over morph identities. Trials from images (not including 50% morphed images) in 9 (monkey T) or 18 (monkey X) morph identities were used for network training. For each image used for training, 6 trials (monkey T, minimal trial number across images and days) or 9 trials (monkey X) were randomly selected and split into two sets: training set (T: 5 trials; X: 8 trials) and validation set (1 trial for both monkeys). Six randomly selected trials (monkey T) or 9 trials (monkey X) of each image in the left-out morph identities were used for testing the trained network. We repeated the 10-fold cross-valida-tion 100 times over different random selections of neurons and trials. In the shuffled decoding, category labels in training trials were ran-domly shuffled while category labels in test trials were unchanged. The differences of two mean decoding accuracies were determined as significant if the difference was located out of 95% confidence interval of 1000 permutation distributions. When comparisons spanned mul-tiple days, *p* values were corrected for multiple testing using the Benjamini-Hochberg false discovery rate (FDR) procedure. To deter-mine whether the correlations between category representations in one brain area and day number were significant, we compared the correlations of decoding accuracy vs. day number and shuffled decoding accuracy vs. day number with bootstrapping tests. For each bootstrapping sampling (5000 times in total), decoding accuracies or shuffled decoding accuracies were sampled with replacement and averaged for each day. Then, Pearson correlation against day number was computed across learning days. If the 95% CI of the correlation coefficients differences distribution did not include 0, decoding accuracies were significantly correlated with day number.

To compare the time course of the monkeys' behavioral perfor-mance with decoding accuracy, we adopted three methods. First, we performed a shuffled control analysis by comparing the correlation between decoding accuracy and behavioral performance to the cor-relation obtained using shuffled decoding accuracy, with significance assessed via bootstrapping. Second, we directly compared the average decoding accuracy in areas TE and TEO with behavioral performance across days using a linear regression model, with decoding accuracy as

the regressor and behavioral performance as the dependent variable. Data from both monkeys were pooled to increase statistical power, and monkey identity was included as a fixed categorical factor to account for individual differences in task design and learning dynam-ics. To assess whether the strength of the relationship with behavior differed between TE and TEO, non-parametric bootstrapping correla-tion analysis was performed. Third, we quantified the unique associa-tion between TE or TEO population signals and behavior using partial correlations. Specifically, we computed Pearson partial correlations between TE decoding accuracy and behavioral performance while partialling out TEO decoding and monkey identity (0 = monkey T, 1 = monkey X). The reciprocal analysis tested TEO vs. behavior con-trolling for TE and monkey identity.

## Choice probability and choice decoding

Choice probability was computed for each neuron using ROC analysis. Trials were grouped based on the animal's choice (i.e., bar-release in the first vs. second interval), and for each neuron, we calculated the area under the ROC curve (AUC) to quantify how well firing rates dis-criminated between choices. The second-interval choice was designated as the positive class. To enable direct comparison of choice selectivity across neurons, AUC values < 0.5 (favoring first-interval choices) were transformed to $1 - AUC$, such that all AUCs >0.5 reflected choice dis-crimination strength regardless of direction. We assessed significance using permutation tests by shuffling choice labels across trials.

To train the choice decoder (using the same procedure described in the "Category decoding in the neural population" section), trials were grouped according to the animal's choice, regardless of image identity. For each choice condition, trials were randomly divided into training (80%), validation (10%), and test (10%) sets. To assess chance-level performance, choice labels in the training set were randomly shuffled. This decoding process was repeated 500 times for each learning day.

## Category decoding in correct trials and error trials

To measure the decoding accuracy in correct and error trials and compare them across learning days, we faced the issue that trials for each image are limited, and the number of correct and error trials for each individual image varies substantially across learning days. We therefore adopted the following approach which allowed us to com-pare decoding accuracy in correct and error trials with sufficient sta-tistical power. For the training procedure, only the correct trials from images with 0, 25, 75 and 100% morph level were used. These images had enough correct trials to train the decoder reliably. Only the images with at least 3 (monkey T) or 6 (monkey X) correct trials across learning days were included. To balance the numbers of cat and dog images fed to the decoder during training, if one cat (or dog) image was excluded, the dog (or cat) image of the same morph identity with the corre-sponding morph level (0% vs. 100%, 25% vs. 75%) was also excluded. Once one image was excluded in 1 day, it would be excluded from all days. To include enough images for training and therefore obtain a better category model, we only trained the model using the days when the monkey's performance was not less than 75% (the last 4 days for both monkeys). For the test on correct and error trials, the ambiguous images (morph level: 35%, 40%, 45%, 55%, 60%, 65%) were used. Only the ambiguous images with both correct trials and error trials were included for test, which excluded the potential confounds that correct and error trials were from the different images. At first, we excluded an image across all days if that image did not have correct trials or error trials in 1 day. But this procedure excluded ~ 83% (100/120) and 50% (30/60) of the ambiguous images in monkey X and T, respectively. In order to gain enough statistical power, we excluded images within single days instead of across all days.

We used the same decoders as described in the section "Category decoding in the neural population". We trained the traditional

category decoder with 3 randomly selected (monkey T) or 6 (monkey X) trials from each image with morph level 0%, 25%, 75% and 100%, and tested it with correct and error trials from the valid ambiguous images. This procedure was repeated 200 times. For the generalization category decoder, we used 10-fold cross-validation over morph identities. We tested the correct and error trials from the ambiguous images within the left-out morph identities. The cross-validation procedure was repeated 200 times.

To determine whether the decoding accuracy difference between correct and error trials significantly increased across learning days, we compared the correlations of decoding accuracy difference vs. day number and shuffled decoding accuracy difference vs. day number with bootstrapping tests, similar to the method described in the "Category decoding in the neural population" section.

### Quantifying discriminability of neural ensembles with d′

To quantify category discriminability of neural population responses, we computed the discriminability index ($d'$) from decoder output scores for TE and TEO ensembles individually and in combination. For each decoding repeat (200 repeats in total), $d'$ was calculated as follows:

$$d' = \frac{|\mu_{\text{cat}} - \mu_{\text{dog}}|}{\sqrt{0.5(\sigma^2_{\text{cat}} + \sigma^2_{\text{dog}})}} \tag{4}$$

$\mu_{\text{cat}}$ and $\mu_{\text{dog}}$ are the means of the decoder output scores for test trials from two different categories (e.g., cat vs. dog). $\sigma^2_{\text{cat}}$ and $\sigma^2_{\text{dog}}$ are the variances of those scores within each category.

For individual regions, decoding followed the same procedure as described above. For combined analyses, we randomly selected and pooled the same number of neurons from TE and TEO as used in individual decoding (Monkey T: 8 neurons from each region; Monkey X: 59 neurons from each region). Interpretation of $d'$ followed the following logic: $d'(\text{TE} + \text{TEO}) \approx \max[d'(\text{TE}), d'(\text{TEO})]$ suggests redundant or overlapping information. $d'(\text{TE} + \text{TEO}) > \max[d'(\text{TE}), d'(\text{TEO})]$ indicates complementary, partially independent contributions. $d'(\text{TE} + \text{TEO}) \approx d'(\text{TE}) + d'(\text{TEO})$ reflects fully independent, non-overlapping information. $d'(\text{TE} + \text{TEO}) < \max[d'(\text{TE}), d'(\text{TEO})]$ implies that one region may introduce noise or task-irrelevant signals that reduce discriminability.

### Reporting summary

Further information on research design is available in the Nature Portfolio Reporting Summary linked to this article.

## Data availability

The source data are provided in the Source Data file available at: https://doi.org/10.6084/m9.figshare.29896067. Source data are provided with this paper.

## Code availability

The core analysis code is available at: https://doi.org/10.6084/m9.figshare.29896067.

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

## Acknowledgements

We thank the staff of the Veterinary Medicine and Resources Branch and the Central Animal Facility, NIMH, for animal care and support. We thank the staff of the Section on Instrumentation, NIMH, for comprehensive engineering support. The Japanese monkey was provided by the NBRP-Nihonzaru, which is part of the National Bio-Resource Project of the Ministry of Education, Culture, Sports, Science and Technology (MEXT, Japan). This work was supported by the Intramural Research Programs of the National Institute of Mental Health (project ZIAMH 002032), New Energy and Industrial Technology Development Organization (NEDO), and KAKENHI (19K12149; 20K12588; 22K12189; 19J40302). This research was supported in part by the Intramural Research Program of the National Institutes of Health (NIH). The contributions of the NIH author(s) were made as part of their official duties as NIH federal employees, are in compliance with agency policy requirements, and are considered Works of the United States Government. However, the findings and conclusions presented in this paper are those of the author(s) and do not necessarily reflect the views of the NIH or the U.S. Department of Health and Human Services.

## Author contributions

M.A.G.E., B.L., B.J.R., N.M. and Y.S.-M. designed the experiments. M.A.G.E., B.L. and R.C.S. conducted the surgeries. B.L., N.M., K.M.G., and K.H. collected data. W.W. conducted data analysis. W.W., M.A.G.E., B.L. and B.J.R. wrote the first draft; all authors edited the paper.

## Competing interests

The authors declare no competing interests.
