## [Transparent Peer Review file · Nature Communications]

Enhanced neural plasticity in monkey TE compared to TEO during learning of a feature-ambiguous visual categorization task

Corresponding Author: Dr Mark Eldridge

Version 0:

Reviewer comments:

Reviewer #1

(Remarks to the Author)

The authors trained 2 monkeys (one is Japan and one in the US) to discriminate dog and cat images (10 morphed pairs) and recorded during the training single neurons in both TEO and TE using Utah arrays. The main result of this major effort was an improvement of the category representation in TE but not in TEO, while lesions of these areas had only modest effects (in a previous study).

The main weakness of the study is an asymmetric result, as in both monkeys only the dog representation improved, not the cat representation. Thus the general claim the authors want to make seems not to be supported.

Furthermore, there are disturbing differences between the two monkeys, in particular in figs 2 and 3. Even in fig 4 which almost convinced me, there still is a difference between monkeys in the dog generalization in TE. Thus the effects of learning and reward schedule are not completely dissociated.

Since this is not the first report of changes in TE representation of shape/categories with learning, these weaknesses greatly reduce the importance of the present manuscript. It seems that to reach a firm conclusion, at least on more monkey may be needed, either with two different categories, or with dog and cat presented in reversed order to favor cat images more. Using neuropixel probes, which yield more neurons than the older Utah arrays might yield stronger data.

The dog representation improved with training which by definition increases the number of correct trials. Thus not surprisingly the decoding of category in TE also was improved in correct vs error trials (fig 5). While this may seem trivial, it speaks to the degree to which the category learning reflects the TE changes, a point not explicitly addressed. But both fig 3 and the accompanying S figures suggest the neuronal changes in TE might be modest compared to the behavioral changes. It might be worthwhile to test this formally.

Minor points:

- 1 the definition of TEO should use monkey specific reports eg the old Ungerleider study of the more recent Kolster et al 2014 fMRI study;
- 2 the discussion is rather flat.

Guy Orban

Reviewer #2

(Remarks to the Author)

This paper describes a category learning experiment in two monkeys, in which neural recordings were taken in areas TE and TEO of the ventral visual pathway. The category learning task used photographic stimuli created from morph sequences which blend from a cat at one extreme to a dog at the other. Monkeys learned to discriminate these morphed items as “cat-like” or “dog-like” using an unusual temporal window-dependent lever-release task that these authors have used in several previous studies. The authors claim to have found neural plasticity in TE but not TEO in response to category learning. Specifically, in the TE neuronal population only, category specificity and category generalization (as measured by two kinds of category decoding classifier) increased with task learning, and category decoding accuracy was enhanced in correct trials

relative to error trials. Most of these effects were not found in the TEO neuronal population, and in a couple of analyses, the category information present in TE was found to be statistically greater than that in TEO. (But I have substantial concerns about the claims of differential category *learning* in TE vs TEO, owing to a lack of appropriate statistical tests, see below).

This study makes a valuable contribution to literature, although the task is not novel. In addition, one could argue that the findings are rather incremental: prior studies have investigated the learning of category representations in IT (e.g., Baker, Behrmann, Olson, 2002; De Baene, Ons, Wagemans & Vogels, 2014; Pearl et al. 2024 from this same group), in similar ways. The novel advances here are that the authors track the learning during training (rather than measuring it before and after, which leads to similar conclusions) and that they compare the specific subregions TE and TEO (but, again, I have substantial reservations about their claims of differences between TE and TEO, described below).

Although the task is a bit odd and asymmetric (e.g., for Monkey M, only dogs are rewarded, cats never are, and for Monkey T, cats must be responded to in the earlier time window and result in a smaller reward than dogs, which must be responded to in a later time window and yield a slightly larger reward), the behavioral results seem to mitigate most concerns. Specifically, there is no response bias (toward dogs) in Monkey M after learning. In addition, the RT data make sense in terms of the task contingencies and imply that monkeys are able to recognize both kinds of stimulus (cat/dog). For example, it took Monkey M longer (more days) than Monkey T to learn to respond quickly in the first interval if it's a cat image, which makes sense because Monkey M had no reward for cats (whereas Monkey T did), but Monkey M did eventually learn to respond more quickly to cats because that enabled him to end this non-reward trial more quickly and move on to a new trial with a chance of reward. In contrast, Monkey T actually got a reward for responding quickly to a cat image in the first interval, and thus he learned to respond quickly in the first interval on an earlier day in training.

However, I have two major concerns. The first concerns the lack of appropriate statistics supporting the assertions made about differences in *learning* between TE and TEO. The second is a lack of clarity on some of the methods which must be rectified before a proper assessment of these results can be made.

Appropriate Statistics. A major claim of the article seems to be that area TE does things (category learning things) that TEO does not. That there is a *difference*, functionally, between TE and TEO when it comes to category *learning*. This conclusion hinges upon the finding that whatever change occurs pre- to post-learning in TE is a larger change than any change happening in TEO. And yet the authors never perform a statistical test for this claim; specifically, a statistical test that measures an interaction between brain region (TE, TEO) and stage-of-learning (day number). In almost all of the analyses, they simply demonstrate, *separately*, that TE shows some neural plasticity effect and TEO does not. (Or that TE holds more category knowledge than TEO without testing for the *change* in category knowledge.) To claim a difference in category learning between TE and TEO on these grounds is statistical error 101!! Remember in undergraduate stats when they told us "the difference between a significant difference (e.g., $p=0.049$) and a non-significant difference (e.g., $p = 0.051$) is not necessarily significant"? This is the exact error being committed here. Remember that with null hypothesis significance testing, one cannot conclude that there is no category learning in TEO, only that there is a lack of evidence for learning in TEO (these are quite different). See also Nieuwenhuis et al. (2011) Nature Neuroscience.

Here I provide a non-exhaustive list of the places where a statistical comparison of the change in TE vs change in TEO should have been made, but was not:

- Fig. 2D – "percentage of neurons modulated by visual category" vs. "day number" (correlation) was computed separately for TE and TEO but the size of the correlation in the two brain regions were not compared (e.g., some test of the interaction between brain region and day number; there may not exist a common statistical test for this, but the authors could get creative and at a minimum do some kind of non-parametric bootstrapping test on the differences between TE and TEO on different days)
- Fig. 2E – visual category effect size vs day number, compared separately in TE and TEO, but the difference between these effects in TE vs TEO was not tested for significance (i.e., some test of the interaction between brain region and day number)
- Fig. 3E,F and 3G,H: no statistical comparison of TE vs TEO was made
- Fig 4 – TE and TEO were compared in this analysis! However, as far as I could tell there was still no test of the interaction between brain region and day number, necessary for a demonstration of *learning* in TE that does not occur in TEO.
- Figure 5, as for Fig. 4: some comparison of TE vs TEO, but no test of the interaction with day number.
- More of the same in the supplemental figures

To be clear: even if the authors found no interaction between brain region (TE, TEO) and day number, this would not mean the results are worthless. As the authors themselves conclude, maybe TEO is doing the same thing as TE but a bit more slowly. HOWEVER, this would necessitate rewording of the conclusions drawn: the verbal descriptions of the conclusions currently in many prominent places imply that there is a contrast between TE and TEO and that this is functionally significant. If these claims are to be retained then the authors need to do the correct statistical tests to back them up.

Lack of clarity. There are points where the methods are opaque. Most importantly: How were the neural AUC curves calculated? What feature of the experimental design or data provided the threshold settings for plotting the ROCs? (The morph level of the stimuli?) Are the "True Positive Rate" and "False Positive Rate" referring to the monkeys' behavior? (As in human recognition memory ROC curves?) Or are they referring to some measure of neural firing? The method for calculating these neural ROCs is never explained and the reader cannot be expected to guess, read other papers, or know in advance.

Other places where there is a lack of clarity:

- Figure 2C: the figure and caption are very unclear. Where are the image onset, red square onset, and possible response window onset in the timeline? Why does the monkey appear to be releasing the bar at approximately the same time for all but the most doglike of images? Why aren't the bar release times delayed, as the morph level increases (and images are

more catlike)? Why does it appear that even for the cat-like images (morph levels under 50%), monkeys were releasing the bar in the same interval as for the dog-like images?

- Figure S1B, are the RTs for the dog-like images measured with respect to the start of the SECOND interval? (this is not stated).
- Figure 2E – the calculation of the “Effect size of visual category” needs better explaining
- A better definition (appearing earlier) is needed for “morph identity”. It took a long time for me to realize that this refers to one specific morph sequence with a unique pair of dog-cat parents. Clarify earlier.
- Figures 3 and 4: clarify explicitly whether asterisks indicate test of significance between TE vs TEO or between day 1 and day 9 within a given brain region
- Some of the methods need editing by a native English speaker – grammatical errors
- Lines 328 to 330: text implies that the optimal response strategy was simply to always release the bar as soon as the square turned green (never during the red square). This is obviously not quite correct. It depended whether the image was cat-like or dog-like (if cat-like it makes sense to release early to move on to the next trial, but if dog-like it's optimal to wait). Explain this here so that the reader is not confused.
- Lines 420 – 421 unclear which images were being used. Please edit for clarity.

Reviewer #3

(Remarks to the Author)

This study addresses an interesting question and was exciting to read. It is well written, with good design elements and comprehensive set of analyses, and several of the findings are compelling. The data show that TE category representations increase over the course of learning, tracking performance both over time and on trial-by-trial basis. My main concerns regard robustness, lack of nuance in treating the results, and potential limitations of conclusions driven by the experimental design. I would also like to get a better idea of the authors think about the significance of this TE/TEO results on a mechanistic level. Let me clarify those in more detail.

1. Robustness. Some of the key conclusions hinge on results that are based on as few as 8 TE neurons in Monkey T. The noisiness of that data is apparent in several figures. Some of the key conclusions hinge on days 7-9 in Monkey T, and would have been different if Monkey T would only have 5 days or recording like Monkey M had. Some of results can only be seen for dog but not cat images, even in monkey where cat and dog categories are supposed to be balanced. All of this makes me less confident in the robustness of the results.

I understand the challenges of obtaining data like this, but more could be done to acknowledge those limitations (beyond the asymmetry for cats/dogs discussed currently). My concerns are eased somewhat by the population coding analyses that I see helping somewhat with the small-N TE neuron count in Monkey T. The authors could consider starting their results with those, to establish the main pattern of findings, and then provide the deeper dive focusing on single cell-based analyses that are interesting and informative, but need to a little more careful framing.

2. Lack of nuance in interpreting results. Eventually, as a whole, I mostly accepted the key conclusion of the paper, especially as it drew from the performance-based analyses (link to behavior) and changes over time. The abstract does a good job making clear what key findings are drawn upon for conclusion. However, conclusions of several individual analyses don't seem justified and the data are painting a potentially interesting picture that is never mentioned. Examples:

- Lines 103-110 note that category selectivity was similar between TE and TEO, but TEO had greater mixed selectivity, so that means that TE is the category selective one. That's not necessarily an obvious conclusion.
- Figure 3 related analyses emphasize that category separation increases over training in TE. But a closer look suggests that category separation already starts high in TEO, so TE is just catching up in category representation that is already starting strong in TEO. Then, the results are not holding up for cat in Monkey T (where cat/dog are both rewarded). And in Monkey M, TEO does show greater AUC (category separation) over TE even for cat, across all days. The conclusions don't really address all these nuances, or the fact that TEO is showing strong category separation early. Given prior lesion findings that both TE and TEO contribute to category learning and potentially interact (largest effect when both are lesioned) could be maybe linked to this finding of early category representations in TEO. Even if it's not currently clear how the regions may interact, this piece of data may prove valuable to our understanding eventually.
- Similar issue/nuance exists in Category decoder analysis, where category decoding in both monkeys starts stronger in TEO than TE. See also my comments on design where the early TEO category representation could come from. While the abstract is careful in verbalizing the findings, the title may be a little overreaching given the existence of category representations in TEO that may precede category representation in TE. I understand the argument of behavioral relevance but we cannot rule out TEO contributions to category learning given the design and data.

3. Design. There are a couple of aspects of the design that need to be emphasized more prominently than they currently are: the different procedures for the two monkeys and the pretraining regimen.

- Different procedures. I understand that the difference was not intended to be a manipulation in its own regard. Nevertheless, the difference between one-category learning (A/notA) and two-category discrimination (A/B) is an area of its own investigation, as least in human primate literature. While the tasks could be in principle treated the same, empirical research in humans suggests otherwise. For example, A/notA can be learned by amnesiacs and recruits striatum and perceptual learning system more heavily while A/B cannot be learned by amnesiacs and recruits hippocampus more heavily. Thus, for broader impact on the field, it's important that this paper makes it more apparent that both of these category

learning tasks were used, A/B in one monkey and A/notA in another. While I understand this may not make it into the abstract, a more explicit notion in the text (plus a link to the human literature that dissociates them) is needed. Figure 1A should be also edited to make it clear the design only applies to one Monkey and what was the difference for the other monkey. Rationale for this difference should also be offered: was it intentional? Accidental? It is exciting to see the results that were consistent across monkeys. It may also be fruitful in the future to consider some of the inconsistent findings in light of these procedural differences, even though they are currently not attributable to the procedure difference based on a single monkey undergoing each procedure.

- Extensive pre-training. As the authors noted, this study would not be possible without extensive pretraining as trainings can take weeks to months. However, it is important to mention the extensive pre-training prior to methods, and consider it during interpretation/discussion of the results. We do not know how these regions contributed to that initial learning, so it may have an important implication for the findings. It's maybe a design aspect responsible for the strong category representations existing in TEO already on day 1, which may require further nuance in the interpretation/discussion.

4. Big picture. The general discussion was somewhat limited in scope, discussing one limitation (there are more not mentioned) and not offering much in terms of significance, or how the circuitry may work mechanistically. Can any more thoughts be offered? Why TE rather than TEO when we focus on changes over time and link to behavior?

5. Smaller comments

- Figures were generally of high quality. Figure 1 should have the two different procedures visualized or somehow noted, given my prior comment. Figure 7S may be more useful is some other visualization (maybe line graph) given that displays decoding accuracy with chance of 50%. The bargraph starting at zero makes it difficult to see any deviations from chance, and the line graph still allows visualization of differences between conditions.
- In general, it would be helpful to provide more rationale for some of the design choices and some explanations.
- Why were there so few TE neurons sortable from two arrays in Monkey T, while substantially more were captured by one array in TEO in that same monkey, or even more by the arrays in the other monkey?

Reviewer #4

(Remarks to the Author)

In this manuscript, the authors use chronic recording arrays in two areas of macaque temporal cortex (TE and TEO) in combination with relatively rapid category learning, to try to establish the nature and distribution of signals that may support the establishment of new categories. The question is very significant: while there are many theories and data about category tuning (including some human fMRI studies), there is far less data on the emergence of categories through learning, and in particular whether there are differences between TE and TEO. Recent lesion studies, cited in the introduction, point to a parallel processing of signals supporting categorization between the two areas, and the authors wanted to examine this during the actual process of learning, in which the complications of interpreting lesions data (such as targeting and compensatory mechanisms) are eliminated. The authors have an impressive data set, appropriate analyses at both the single cell at ensemble level and some nice behavioral evidence of learning, but inconsistencies between the animals, as well imprecise language with regard to categorization computation (as opposed to attribute encoding), detract from its potential impact.

Specific Comments:

1. The task design, with its temporal structure, certainly makes for numerous potential confounds, including pre-motor, reward anticipation, and passage of time encoding, and by in large the authors do a good job of trying to address many of these in the discussion. But there were training differences between the animals in terms of the task, and also several instances, where the animals' physiology was not consistent, and a concise discussion of whether the two (training and physiology differences) could be related is warranted. especially since there are numerous differences between animals. Examples include morph identity neurons (Fig S3), category selectivity (Fig S5B&E), and error effects (Fig S7B). These differences can have major implications: looking at S7 for example, it is clear that TEO "conclusion" (that it doesn't have performance related information) relies on a single animal. In general, given all these differences, the authors need to be much more circumspect in their statements. For example, specifically with regard to the error analysis, it is concluded that TEO is not a "plausible candidate region for the computation of category membership." How does Monkey M feel about that?!

2. Speaking of computing categories, I think the authors are on dangerous ground with such statements. As demonstrated by Kobatake and Sigala, attribute tuning is subject to learning, and with population decoders you can get category performance even from signals selected solely according to their shape preferences (Bougou et al, 2024). So if a region's attribute tuning changes that helps categorization, but does not imply that categorization is computed in that region: any task in which that attribute is relevant (discrimination, detection in noise) would also be helped. Looking at the few examples in individual cells that are presented (more on that later), I really don't see anything that I would consider a category cell (that fires the same across morph identity and only varies in morph strength).

3. In this regard, it would be nice to look at, in both individual neurons and a population level, trial-by-trial choice effects for the 50% (even near 50%) morphs. While positive choice probability is challenging to interpret (attention vs read-out), the lack of any choice signals would seem to constrain how strongly these signals contribute to the actual categorical choices made by the animals.

4. The authors do a good job of testing the decoders within and across categories with the dissimilarity metric (Fig 5E). But what about consistency with Fig S1A: shouldn't the decoders be much worse at morphs near 50% than they are ones that aren't?

5. The authors have a lot of data, but in some cases only examples are shown where I think a summary could be shown as well. For example, with regard to morph identity/level tuning (Fig 2C), why not, as a function of training (or even first day vs last day), show a population summary where morph identity and morph level are sorted for each neuron, normalized to the peak response, and averaged? It would be nice to know if morph identity tuning is becoming broader and morph level tuning more binary (both consistent with categorization) with learning. I'm also not sure where the population decoding neuronal sample size in Fig 4 comes from: are the 8 and 59 neurons those neurons which are thought to be present in every single day's recordings? How was neuronal identity across days verified? And what is the population in TE and TEO respectively? What happens if I just through every neuron recorded in the few days and compare that with every neurons recorded in the last few days? And is there any chance that decoding differences between the animals (for example, the lack of within dog similarity in Monkey T Fig5E) are simply constrained by smaller sample sizes?

6. The manuscript starts with the conclusion based on lesion studies that there is a "parallel" system for categorization between TE and TEO, but this point seems inconsistent with how the discussion and abstract are framed, which strongly emphasize TE. But could the parallel scheme be tested by computing d' for TE and TEO ensembles separately, and then together, and seeing if the together d' is consistent with independent detectors?

7. TE has been divided in to TEpd (dorsal posterior) and TEad (dorsal anterior) on the basis on connectivity (for example Saleem et al, 2000, Kravitz et al, 2013) and it looks the authors' have arrays in both. Any differences?

Version 1:

Reviewer comments:

Reviewer #1

(Remarks to the Author)

the authors have adequately answered my comments, and those of the other reviewers;
the collection of novel data is especially wellcome

Reviewer #2

(Remarks to the Author)

In general, the authors have done a good job addressing my concerns. The new analyses examining the interaction of brain region by day have greatly improved the manuscript and in most cases the authors' claims of enhanced learning are now justified. There have also been many clarifications which improve the manuscript's readability. I have just a few remaining minor concerns.

On at least two occasions, the "brain region x day" interaction was significant in one animal but not the other. This is somewhat glossed over, as though the result were found in both animals:

- On page 4: "In monkey T, the proportion-versus-day correlation was significantly stronger in TE than in TEO, whereas in monkey X the same trend did not reach significance" but later in the paragraph "Thus, it appears that category-modulated neural activity in TE, but not TEO, increased during learning"

- On page 9: "which indicated that encoding of one category (dog images) was enhanced in TE during learning"  only true in one of two animals.

- In both cases, the authors need to state in the final conclusion, not just buried in the results part, that it was only found in 1 of 2 animals.

- Needs to be mentioned again on page 15 where inter-animal differences are listed.

Relatedly, on page 16, where it is stated "our key finding — enhanced neural plasticity in TE compared to TEO during learning of our ambiguous visual categorization task — was consistent across both subjects" needs to be tempered/watered down. It was not *consistently* found because in at least two cases, the effect of *learning* was found in only 1 of 2 animals. I appreciate that it was "often" found, but I think the wording needs to be made a little more circumspect. Perhaps "typically found", or "found in most analyses" or similar?

Figure 1A (the experimental task) is very unclear. It looks like Monkey X (denoted by dashed lines) only has the option to do the same thing for both cat-like and dog-like, and in either case the only outcome is to go to the next trial. This cannot be right. It contradicts the description in the methods: "The optimal response strategy for monkey X depended on the visual stimulus... For cat-like images (which were never rewarded), releasing early allowed the monkey to efficiently skip the non-rewarding trial. In contrast, for dog-like images (which could lead to reward), it was advantageous to wait until the green square (second interval) before releasing the bar to obtain a reward." I suggest revising this figure.

Line 576: "increasement" (not an English word)  "increase"

Overall, this work will now make a valuable contribution to the literature.

Reviewer #3

(Remarks to the Author)

The authors did a good job addressing many of my and other reviewers' comments. While more data would have been ideal, I recognize that science is incremental and cumulative, and the current paper makes a sufficient contribution within the scope of the data available. I just suggest to acknowledge the category signals in TEO (pre-existing those in TE) in the abstract, in addition to acknowledging them in text. As discussed, all the TEO effects could have happen during pre-training and not be observed here for that reason. The abstract, as written, may provide a somewhat misleading impression regarding the role of TEO.

Reviewer #4

(Remarks to the Author)

I appreciate and applaud the rigor with which the authors have added additional analyses and modified their conclusions, but, especially with consideration of these new additions, do not find the central tenet, as specified in the final line of the abstract, that TE plays a "more fundamental role" in category learning, well justified by the data. Indeed, the change in the title, which now talks about plasticity during learning, rather than explaining learning, makes me wonder if the authors still feel strongly about this claim given their data. This is a complicated data set, but the inconsistencies between analyses and animals are just too prevalent.

1. There are many differences between the animals physiologically: I identified a lot of them in the previous review and they are many more that appear now in the new supplementary figures. But perhaps more fundamental for a study trying to understand the physiological underpinning of learning are the data suggesting that the animals learned different things with different dynamics. We can see this in the purely behavioral data of Fig 1 and S1. In Figure 1, Monkey T learns gradually and at an almost linear rate: if you define learning as a change in performance with training, you would say that the learning (the slope) is relatively constant throughout training. By contrast, Monkey X, has a extremely non-linear curve, and clearly the greatest change happened between days 1 and 2. This would be consistent with a pure task familiarity effect: in the first day, he wasn't consistently sure what the task actually was, but as soon as he realized it (by day 2, and possibly in the later half of day 1), his performance improved substantially and had modest improvement after that. This is not really category learning in the sense of refining sensory representations, rather its task learning in trying understand the relationship between sensory representations and reward. Figure S1 supports this: Monkey T increased his sensitivity systematically (the slope of the psychometric), while Monkey X largely did not: he simply changed his criterion and systematically reduced bias (especially in the first day) without obvious changes in sensitivity. Both bias reduction and sensitivity improvement result in behavior improvement but conceptually (e.g. standard decision theory), and potentially physiologically, are completely different. So we have N=1 for improved category sensitivity, and N=1 for changes in criteria, but not N=2 for either one.

2. Largely based on the previous sets of comments, the authors commendably added a number of analyses, most of which appear in the form of supplementary material. Problematically, I don't find these results very consistent or supportive with the "TE is responsible for learning" claim. Evidence in support of that claim should show that the time course of changes in TE is a significantly better match than changes in TEO to explain both animals' learning dynamics. By simple visual inspection, I would say almost all of the main body figures, with the exception of 3E, are supportive (although I do have statistical concerns which are stated below). However, if I make a check list of supportive data in the supplementary materials that focus on population level analyses, things are a lot less clear. In Figure S3, B is supportive, but A,C, and D are not. Neither S5A or S5B seems supportive. In figure S6 A and C are, but B and D are not. And finally, I don't see anything in S8, S9 or S11 that is particularly supportive. I'm particularly concerned about the choice analyses: if signals in these regions are not consistently predictive of choices, isn't making the argument that these signals are responsible for better choices dangerous? Doesn't it make the biological relevance of "ideal decoders" less compelling?

3. So if TE isn't solely responsible, is it more responsible than TEO? Here I have to agree with other reviewer that if A is significant and B is not, the difference isn't necessarily significant. There are three approaches that come to mind here (but I'm sure there are others): one is to specifically look for synergies or redundancies (which the d' does), the other would to try do some partial correlation analysis (time course of TE vs learning, given TEO and TEO vs learning given TE), or a regression approach in which the factors and their interactions are all incorporated: $aTE + bTEO + cTE, TEO = \text{learning}$. An additional point here, with regard to synergies/redundancies, is a lost opportunity: if I combine simultaneously recorded TE and TEO neurons into one large population basket, how does it do and how does it compare with TE and TEO separately? Taking the classic signal vs noise correlation approach, if signal correlations between TE and TEO are in a different direction than noise correlations, then you should be able to decode much better with both populations than with either one individually. I think this is what is going on in S9 (which really should be merged with 3B and D in the main text), but why not do this for the rest of the analyses of Figures 3 and 4? And, for the question of learning, does the dynamics of the combined population better match behavioral changes than the dynamics of the individual populations?

Version 2:

Reviewer comments:

Reviewer #4

(Remarks to the Author)

The authors have done an outstanding job addressing my concerns with language, adding new analyses (including TE and TEO pools) and partial correlation analyses. To be honest, it's probably the most thorough response I have ever seen, so kudos for a great paper.

We thank the reviewers for their thoughtful comments and constructive suggestions. We have revised the manuscript accordingly, addressed all concerns in detail, and incorporated the recommended changes throughout the text.

Reviewer #1 (Remarks to the Author):

The authors trained 2 monkeys (one is Japan and one in the US) to discriminate dog and cat images (10 morphed pairs) and recorded during the training single neurons in both TEO and TE using Utah arrays. The main result of this major effort was an improvement of the category representation in TE but not in TEO, while lesions of these areas had only modest effects (in a previous study).

The main weakness of the study is an asymmetric result, as in both monkeys only the dog representation improved, not the cat representation. Thus the general claim the authors want to make seems not to be supported.

We thank the reviewers for the insightful comments. Although the cat representation did not show a clear increase during learning in the single-neuron analysis—which focused on increased neural responses to cat-like versus dog-like images (Figure S8F–K)—we did observe representational changes at the population level using representational dissimilarity analysis (Figure 3E, middle). Specifically, for both monkeys, the population response patterns evoked by cat-like images became more consistent over the course of training, suggesting improved internal structure and stability in the neural encoding of these stimuli, even in the absence of increased overall response magnitude.

Furthermore, there are disturbing differences between the two monkeys, in particular in figs 2 and 3. Even in fig 4 which almost convinced me, there still is a difference between monkeys in the dog generalization in TE. Thus the effects of learning and reward schedule are not completely dissociated.

The individual differences between the two monkeys are not unexpected, given variations in task design, species differences, and the unequal number of recorded neurons, with substantially fewer neurons recorded in monkey T, which may limit sensitivity to detect learning-related changes. Despite individual differences, our key finding — that neural plasticity in TE, not TEO, reflects learning in our visual

categorization task — was consistent across both subjects, demonstrating that the core effects generalize across animals despite individual variability.

Although behavioral results mostly mitigate the concerns, as pointed out by reviewer #2 and as discussed in the Discussion section, we still cannot rule out the possibility that learning and reward schedule may not be entirely dissociated.

Together with other reviewers' comments, we now discussed the difference on line 443 – 451: We observed several inter-animal differences including: neurons modulated by morph level/identity (Figure S3), category selectivity neurons (Figure S8B&E), dog generalization (Figure 3E), and error effects for TEO (Fig S6B&D). These differences may stem from differences in task design (e.g., reward contingencies), species differences, and/or the unequal number of recorded neurons - substantially fewer neurons were recorded from monkey T, which limits our statistical power to detect learning-related changes. Despite these individual differences, our key finding — enhanced neural plasticity in TE compared to TEO during learning of our ambiguous visual categorization task — was consistent across both subjects, demonstrating that the core effects generalize across animals despite individual variability.

Since this is not the first report of changes in TE representation of shape/categories with learning, these weaknesses greatly reduce the importance of the present manuscript. It seems that to reach a firm conclusion, at least on more monkey may be needed, either with two different categories, or with dog and cat presented in reversed order to favor cat images more. Using neuropixel probes, which yield more neurons than the older Utah arrays might yield stronger data.

We did conduct additional categorization tasks with monkey X (For consistency with our previous publication (Pearl JE, *et al.*, 2014) and to facilitate comparison, 'monkey M' has been renamed 'monkey X' in the revised manuscript), including car/truck categorization using 20 cars and 20 trucks (2 days), and 240 cars and 240 trucks (1 day), as well as cat/dog categorization without morphing (20 cats and 20 dogs for 3 days; 240 cats and 240 dogs for 1 day).

We were able to replicate the main findings using the non-morphed cat/dog tasks (Figure S10, left panels). However, we did not observe systematic changes in either TE

or TEO during the car/truck categorization tasks (Figure S10, right panels). This suggests that plasticity in TE is specifically engaged during categorization learning that involves high visual ambiguity and requires fine-grained perceptual discrimination, such as in the case of cat/dog stimuli. In contrast, simpler visual categorizations, like car/truck, which rely more on coarse visual features (e.g., geometry, size, and/or curvilinearity), may be adequately processed by earlier visual areas.

Considering this, we have updated the manuscript title to more accurately reflect the scope of our findings:

Enhanced neural plasticity in monkey TE compared to TEO during learning of a feature-ambiguous visual categorization task

We now added one section in Results on line 267 - 311:

Learning-related plasticity in TE emerges only for feature-ambiguous categorization

To test whether the results described above generalized to other visual categories, we conducted additional categorization tasks with monkey X, including car/truck categorization using 20 cars and 20 trucks (2 days), and 240 cars and 240 trucks (1 day), as well as cat/dog categorization without morphing (20 cats and 20 dogs for 3 days; 240 cats and 240 dogs for 1 day). The task design was the same as Figure 1A for monkey X. Cat or car trials were not rewarded, dog or truck trials were rewarded for a release on green.

We replicated the main findings using the non-morphed 20-cat/20-dog categorization tasks (Figure S10, left panels). Decoding accuracy in area TE, but not TEO, increased across learning sessions (Figure S10D). Critically, decoding accuracy of category decoder in TE—but not in TEO—was positively linked to behavioral performance (bootstrapped Pearson correlation, 95 % CI lower bounds: 0.01 for TE, -0.55 for TEO; brain region \times decoding accuracy interaction: $p = 1.2 \times 10^{-14}$, LM. For the generalization decoder, TE accuracy showed a positive yet non-significant trend: 95 % CI lower bound = -0.06, whereas TEO again exhibited no association: lower bound = -0.48; interaction: $p = 1.2 \times 10^{-38}$). TE and TEO carried complementary and partially independent category information on multiple days (Figure S10E, 4/6 days for both decoders). TE, but not TEO, population response patterns evoked by dog images became more similar over training, while cat-related patterns did not show such improvement in either TE or

TEO (Figure S10F). This asymmetry may reflect the fact that this was the monkey's first exposure to a cat/dog categorization task. Given the task's asymmetric reward structure and the limited training period (3 days), there may not have been sufficient time for representational changes for cat images. Moreover, categorical representations in TE—but not TEO—were enhanced during correct compared to error trials (Figure S10G, H). At the single-neuron level, TE responses became more evenly distributed³³ across both cat and dog images with learning. No evidence of learning-related changes was observed in TEO (Figure S10I).

However, for the 20-car/20-truck categorization task, we failed to detect any systematic changes in either TE or TEO responses across learning (Figure S10, right panels). This likely reflects the lower visual ambiguity of the car/truck stimuli compared to the cat/dog images. Cars and trucks differ along relatively coarse, low-level visual features such as shape and geometric contours, which may be sufficiently processed by earlier visual areas. In contrast, categorizing cats and dogs—especially when morphed—requires fine-grained discrimination of subtle, overlapping features, which require greater demands on high-level visual processing. We also observed TE and TEO carried complementary, partially independent category information on all days (Figure 10N).

We challenged the monkey with a large set of novel images presented in a single day (240-cat/240-dog, 240-car/240-truck). The results were consistent with the above findings. A small but significant increase in decoding accuracy across repeats was observed in TE, but not in TEO, for the cat/dog categorization task (Figure S11C). In contrast, neither TE nor TEO showed any significant increase in decoding accuracy across repeats for the car/truck categorization task (Figure S11G). TE and TEO also carried complementary, partially independent category information in multiple repeats across both tasks (2/5 for cat/dog; 3/5 for car/truck; Figure S11D, H), suggesting complementary contributions to category representation during these repeats. These findings suggest that TE is selectively engaged, and exhibits plasticity, only when the visual categorization demands fine discrimination between perceptually similar and ambiguous stimuli, such as cat and dog images.

The dog representation improved with training which by definition increases the number of correct trials. Thus not surprisingly the decoding of category in TE also was improved in correct vs error trials (fig 5).

If we understand correctly, the reviewer's concern is that correct trials tended to occur later in training, when category representation and neural decoding were stronger, whereas error trials were more common earlier in training, when both were weaker. In this case, even if there were no genuine neural differences between correct and error trials, the overall improvement in neural representation over time could artifactually produce a decoding advantage for correct trials—simply because correct and error trials occurred at different points in training. The decoding analyses in Figure 5 (Fig. 4 in the revised version) were performed day-by-day, meaning correct and error trials were compared within the same day. This approach controls for global changes in representation across days. Therefore, the observed improvement in decoding accuracy for correct vs. error trials across days cannot be attributed to correct trials being overrepresented in later training days.

Within single days, a temporal confound could lead to better decoding accuracy for correct trials if they tended to occur later in the day. To address this, we examined the distribution of correct and error trials within each day. Trials within each day were equally split into 5 sessions.

We found that in some cases, such as monkey T on day 8 and monkey X on days 4 and 5, correct trials were relatively evenly distributed across the session. Importantly, even on these days, we still observed improved decoding of category information for correct versus error trials in TE (Figure 4). Therefore, while a temporal bias toward later-session for correct trials existed on some days, the presence of a decoding difference on days without such a bias suggests that the neural distinction between correct and error trials

was not solely due to their timing within a session. This supports the interpretation that the difference reflects genuine differences in neural representation linked to behavior, rather than an artifact of within-day trial timing.

We hope we have correctly understood and adequately addressed the reviewer's concern.

While this may seem trivial, it speaks to the degree to which the category learning reflects the TE changes, a point not explicitly addressed. But both fig 3 and the accompanying S figures suggest the neuronal changes in TE might be modest compared to the behavioral changes. It might be worthwhile to test this formally.

We thank the reviewer for this insightful comment. We have now used two approaches to compare the time course of the monkeys' behavioral performance with decoding accuracy in area TE.

First, we leveraged a shuffled control analysis: we compared the correlations between decoding accuracy and behavioral performance against correlations obtained using shuffled decoding accuracy, using a bootstrapping test to assess significance.

Second, we directly modeled the relationship between averaged decoding accuracy and behavioral performance using a linear regression analysis. Given the limited statistical power in this approach when analyzing individual animals, we combined data from both monkeys. Monkey identity was included as a fixed effect to account for inter-subject variability, such as learning rate or task variation. For both methods and both decoding strategies, decoding accuracy from TE significantly correlated with behavior performance.

We add one section in results on line 145 - 168:

Behavioral performance correlated with neural representations in TE but not TEO

To assess the relationship between neural and behavioral changes over learning, we used two approaches. First, we performed a shuffled control analysis by comparing the correlations between decoding accuracy and behavioral performance against correlations obtained using shuffled decoding accuracy, with a bootstrapping test to assess significance. Decoding accuracy

of the TE population during learning correlated significantly with monkeys' performance, but that of the TEO population did not (Category decoder: TE: for monkey T, 95% CI of $r_{(TE)} - r_{(shuffle)}$: [0.04 0.82]; for monkey X, 99% CI of $r_{(TE)} - r_{(shuffle)}$: [0.3 1.86]. TEO: for monkey T, 95% CI of $r_{(TEO)} - r_{(shuffle)}$: [-0.68 0.46]; for monkey X, 95% CI of $r_{(TEO)} - r_{(shuffle)}$: [-0.02 1.67]; Generalization category decoder: TE: for monkey T, 99% CI of $r_{(TE)} - r_{(shuffle)}$: [0.04 1.37]; for monkey X, 95% CI of $r_{(TE)} - r_{(shuffle)}$: [0.16 1.67]. TEO: for monkey T, 95% CI of $r_{(TEO)} - r_{(shuffle)}$: [-0.9 0.08]; for monkey X, 95% CI of $r_{(TEO)} - r_{(shuffle)}$: [-0.31 1.01]). Second, we applied a linear regression model directly comparing average decoding accuracy in area TE or TEO with behavioral performance across days. Given the limited statistical power in this approach when analyzing individual animals, we combined data from both monkeys. Decoding accuracy in TE, not TEO, was a significant predictor of behavior for both decoding strategies (Category decoder: $p_{(TE)} = 0.001$, $p_{(TEO)} = 0.41$. Generalization category decoder: $p_{(TE)} = 0.0006$, $p_{(TEO)} = 0.38$). This relationship did not depend on individual subject differences (Category decoder TE: $p_{(decoding_accuracy \times monkey)} = 0.47$, TEO: $p_{(decoding_accuracy \times monkey)} = 0.06$. Generalization category decoder TE: $p_{(decoding_accuracy \times monkey)} = 0.18$, TEO: $p_{(decoding_accuracy \times monkey)} = 0.17$). Furthermore, decoding accuracy in TE was significantly more strongly associated with behavior than in TEO (one-tail 95% CI low bound = 0.07, 0.17 for category decoder and generalization decoder, respectively, bootstrapping Pearson correlation). These results suggest that TE carries behaviorally relevant category information to a greater extent than does TEO.

We added the result in abstract on line 22 -23: Neural representations in TE, but not TEO, correlated with behavioral performance.

We added one paragraph in the method section on line 660 – 670:

To compare the time course of the monkeys' behavioral performance with decoding accuracy, we adopted two methods. First, we performed a shuffled control analysis by comparing the correlation between decoding accuracy and behavioral performance to the correlation obtained using shuffled decoding accuracy, with significance assessed via bootstrapping. Second, we directly compared the average decoding accuracy in areas TE and TEO with behavioral performance across days using a linear regression model, with decoding accuracy as the regressor and behavioral performance as the dependent variable. Data from both monkeys were pooled to increase statistical power, and monkey identity was included as a fixed categorical

factor to account for individual differences in task design and learning dynamics. To assess whether the strength of the relationship with behavior differed between TE and TEO, non-parametric bootstrapping correlation analysis was performed.

We also updated Figure 1C by excluding trials that included 50% morph-level images, directly comparable to the decoding accuracy in Figure 3.

Minor points:

1 the definition of TEO should use monkey specific reports eg the old Ungerleider study of the more recent Kolster et al fMRI study;

We included the papers the reviewer suggested (reference #6 and #8) and rephrased the definition of TEO in the Introduction on line 33 – 36:

In nonhuman primates, IT consists of two subregions, TE and TEO, distinguished by their architectonics and connectivity^{3, 4, 5, 6}. These distinctions are further supported by functional studies⁷ and more recent fMRI investigations detailing retinotopic organization⁸.

2 the discussion is rather flat.

We added discussion on a mechanistic level on line 323 – 341: Mechanistically, we interpret our results as evidence that plasticity in TE supports the formation of more abstract, experience-dependent, behaviorally related category representations. In contrast, TEO may support simpler visual feature processing, which is less plastic, and is hence less influenced by learning.

Based on our findings—that category-related signals in TEO are stronger than those in TE during the early stage of learning (Figure 3B, D, E), that TE alone shows a gradual, learning-related increase in decoding accuracy, we propose that TE and TEO play complementary roles in category learning. Representation of category in both TE and TEO is significantly stronger than chance level (Figure 3B, D). In both monkeys, and across decoding strategies, the category discriminability of pooled TE and TEO neurons—i.e., $d'(TE+TEO)$ —often exceeded that of the stronger individual region (Figure S9, Figure S10E, N, Figure S11D, H). Specifically, at the initial stages of learning, TEO may provide an early, coarse category representation sufficient for basic categorization demands, perhaps based on simple visual features. As learning progresses, TEO, as well as V4 which directly projects to TE¹⁴, continues to supply lower-level visual

feature representations, which are refined and integrated within TE to gradually construct more abstract and behaviorally relevant category representations. This interpretation is consistent with prior lesion studies showing that bilateral removal of either TE or TEO causes only mild deficits, while combined lesions produce severe, long-lasting impairments, highlighting the interdependent roles of these two regions.

On line 354– 364: The greater plasticity of TE compared to TEO during learning may stem from its anatomical position and broader connectivity. TE neurons have larger receptive fields and integrate bottom-up inputs from simple visual features, allowing them to respond to complex stimuli and support more abstract, flexible representations^{14, 34}. In contrast, TEO neurons have smaller receptive fields and respond to simpler features, providing stable encoding of low- to mid-level visual features^{14, 35}. Moreover, TE exhibits broader connectivity than TEO, including reciprocal connections with the medial temporal lobe, orbitofrontal cortex, and ventrolateral prefrontal cortex—regions involved in memory, reward processing, and decision-making¹⁴. These widespread connections may facilitate top-down modulation in response to behavioral relevance, enabling TE to adapt its representations through learning. In contrast, the more limited connectivity of TEO may constrain its capacity for such learning-dependent plasticity.

On line 452 – 457: In summary, our results reveal TE as the principal locus of experience-dependent plasticity that converts ambiguous visual features into behaviorally meaningful categories, whereas TEO provides a stable, relatively low-level feature-based representation. These findings position TE as a key node in the transformation of sensory representations into categorical knowledge, providing important insights into the neural mechanisms by which experience shapes perceptual decision-making.

Reviewer #2 (Remarks to the Author):

This paper describes a category learning experiment in two monkeys, in which neural recordings were taken in areas TE and TEO of the ventral visual pathway. The category learning task used photographic stimuli created from morph sequences which blend

from a cat at one extreme to a dog at the other. Monkeys learned to discriminate these morphed items as “cat-like” or “dog-like” using an unusual temporal window-dependent lever-release task that these authors have used in several previous studies. The authors claim to have found neural plasticity in TE but not TEO in response to category learning. Specifically, in the TE neuronal population only, category specificity and category generalization (as measured by two kinds of category decoding classifier) increased with task learning, and category decoding accuracy was enhanced in correct trials relative to error trials. Most of these effects were not found in the TEO neuronal population, and in a couple of analyses, the category information present in TE was found to be statistically greater than that in TEO. (But I have substantial concerns about the claims of differential category *learning* in TE vs TEO, owing to a lack of appropriate statistical tests, see below).

This study makes a valuable contribution to literature, although the task is not novel. In addition, one could argue that the findings are rather incremental: prior studies have investigated the learning of category representations in IT (e.g., Baker, Behrmann, Olson, 2002; De Baene, Ons, Wagemans & Vogels, 2014; Pearl et al. 2024 from this same group), in similar ways. The novel advances here are that the authors track the learning during training (rather than measuring it before and after, which leads to similar conclusions) and that they compare the specific subregions TE and TEO (but, again, I have substantial reservations about their claims of differences between TE and TEO, described below).

Although the task is a bit odd and asymmetric (e.g., for Monkey M, only dogs are rewarded, cats never are, and for Monkey T, cats must be responded to in the earlier time window and result in a smaller reward than dogs, which must be responded to in a later time window and yield a slightly larger reward), the behavioral results seem to mitigate most concerns. Specifically, there is no response bias (toward dogs) in Monkey M after learning. In addition, the RT data make sense in terms of the task contingencies and imply that monkeys are able to recognize both kinds of stimulus (cat/dog). For example, it took Monkey M longer (more days) than Monkey T to learn to respond

quickly in the first interval if it's a cat image, which makes sense because Monkey M had no reward for cats (whereas Monkey T did), but Monkey M did eventually learn to respond more quickly to cats because that enabled him to end this non-reward trial more quickly and move on to a new trial with a chance of reward. In contrast, Monkey T actually got a reward for responding quickly to a cat image in the first interval, and thus he learned to respond quickly in the first interval on an earlier day in training.

However, I have two major concerns. The first concerns the lack of appropriate statistics supporting the assertions made about differences in *learning* between TE and TEO. The second is a lack of clarity on some of the methods which must be rectified before a proper assessment of these results can be made.

Appropriate Statistics. A major claim of the article seems to be that area TE does things (category learning things) that TEO does not. That there is a *difference*, functionally, between TE and TEO when it comes to category *learning*. This conclusion hinges upon the finding that whatever change occurs pre- to post-learning in TE is a larger change than any change happening in TEO. And yet the authors never perform a statistical test for this claim; specifically, a statistical test that measures an interaction between brain region (TE, TEO) and stage-of-learning (day number). In almost all of the analyses, they simply demonstrate, *separately*, that TE shows some neural plasticity effect and TEO does not. (Or that TE holds more category knowledge than TEO without testing for the *change* in category knowledge.) To claim a difference in category learning between TE and TEO on these grounds is statistical error 101!! Remember in undergraduate stats when they told us “the difference between a significant difference (e.g., $p=0.049$) and a non-significant difference (e.g., $p = 0.051$) is not necessarily significant”? This is the exact error being committed here. Remember that with null hypothesis significance testing, one cannot conclude that there is no category learning in TEO, only that there is a lack of evidence for learning in TEO (these are quite different). See also Nieuwenhuis et al. (2011) Nature Neuroscience.

We thank the reviewer for highlighting this statistical issue. We now have revised the manuscript to explicitly test the interaction between brain region (TE vs. TEO) and day

number across all relevant analyses. These interactions were assessed using linear models with interaction terms and, where appropriate, non-parametric bootstrapping. Where significant interactions were found, they are now clearly indicated in the figures using red asterisks positioned next to a brace that spans the two curves.

We have added methods on line 577– 586: To test whether category learning progressed differently in TE versus TEO, we fitted linear regression models that included a brain region \times day interaction term:

$$Metric_{ij} = \beta_0 + \beta_1 * region_i + \beta_2 * day_j + \beta_3 * (region_i \times day_j) + \varepsilon$$

where $region = 0$ for TEO and 1 for TE. Significant $region \times day$ interactions indicate divergent learning patterns in TE versus TEO, and these are marked in the figures by red asterisks positioned next to the brace that spans the two curves.

For analyses without repeated measurements (e.g. proportion of significant neurons across days, Figure 2E), we compared learning patterns in TE and TEO by bootstrap-resampling the observations and computing the distribution of the difference between their Pearson correlation coefficients.

Here I provide a non-exhaustive list of the places where a statistical comparison of the change in TE vs change in TEO should have been made, but was not:

- Fig. 2D – “percentage of neurons modulated by visual category” vs. “day number” (correlation) was computed separately for TE and TEO but the size of the correlation in the two brain regions were not compared (e.g., some test of the interaction between brain region and day number; there may not exist a common statistical test for this, but the authors could get creative and at a minimum do some kind of non-parametric bootstrapping test on the differences between TE and TEO on different days)

We tested whether the correlation between percentage of neurons and day number in Fig. 2D (as well as Fig S3A, C) is significantly different between TE and TEO using non-parametric bootstrapping correlation test.

- Fig. 2E – visual category effect size vs day number, compared separately in TE and TEO, but the difference between these effects in TE vs TEO was not tested for significance (i.e., some test of the interaction between brain region and day number)

- Fig. 3E,F and 3G,H: no statistical comparison of TE vs TEO was made

- Fig 4 – TE and TEO were compared in this analysis! However, as far as I could tell there was still no test of the interaction between brain region and day number, necessary for a demonstration of *learning* in TE that does not occur in TEO.
- Figure 5, as for Fig. 4: some comparison of TE vs TEO, but no test of the interaction with day number.
- More of the same in the supplemental figures

We tested the interaction between brain region and day number in all the figures using linear regression models with interaction terms (except Fig. S3 A, C, for which the difference between TE and TEO was assessed using non-parametric bootstrapping test as suggested).

To be clear: even if the authors found no interaction between brain region (TE, TEO) and day number, this would not mean the results are worthless. As the authors themselves conclude, maybe TEO is doing the same thing as TE but a bit more slowly. HOWEVER, this would necessitate rewording of the conclusions drawn: the verbal descriptions of the conclusions currently in many prominent places imply that there is a contrast between TE and TEO and that this is functionally significant. If these claims are to be retained then the authors need to do the correct statistical tests to back them up.

Most of the claims were retained except Fig. 2D (monkey X, for consistency with our previous publication (Pearl JE, *et al.*, 2014) and to facilitate comparison, 'monkey M' has been renamed 'monkey X' in the revised manuscript) and Fig. 5F (monkey X). For Fig. 2D (monkey X), correlation of TE and day number tends to be stronger than that of TEO (the lower bound of one-tailed 95% confidence interval is 0). For, Fig. 5F, we did not find significant interaction between brain region and day number for significant dog AUCs for monkey X.

We have made it clear in the results on line 96 – 98: In monkey T, the proportion-versus-day correlation was significantly stronger in TE than in TEO, whereas in monkey X the same trend did not reach significance.

On line 238 – 239: The brain region \times day interaction was significant in monkey T, but not in monkey X.

Lack of clarity. There are points where the methods are opaque. Most importantly: How were the neural AUC curves calculated? What feature of the experimental design or data provided the threshold settings for plotting the ROCs? (The morph level of the stimuli?) Are the “True Positive Rate” and “False Positive Rate” referring to the monkeys’ behavior? (As in human recognition memory ROC curves?) Or are they referring to some measure of neural firing? The method for calculating these neural ROCs is never explained and the reader cannot be expected to guess, read other papers, or know in advance.

We thank the reviewer for this important comment and apologize for the lack of clarity regarding the neural ROC/AUC analysis. The ROC analysis was performed on single neuron firing rates to quantify category selectivity. For each neuron, we compared firing rate distributions for cat-like versus dog-like stimuli (defined based on morph level) and computed the True Positive Rate and False Positive Rate by varying a threshold over firing rates. These values were used to generate the ROC curve, and the area under the curve (AUC) was used as a measure of category discrimination at the neural level. This analysis is entirely based on neural responses and not on behavioral performance.

We have now added a detailed description of how the neural ROC curves were computed in the Methods section on line 590 – 601: For each neuron, we first extracted firing rates during a predefined post-stimulus time window across trials for images within each morph identity. Trials were grouped according to the stimulus category, defined based on the morph level (Figure 1B). Trials with 50% morph level were not included.

We then generated an ROC curve by varying a threshold over the neuron's firing rate distribution and calculating the True Positive Rate (TPR) and False Positive Rate (FPR) at each threshold. Here, TPR reflects the proportion of "dog-like" trials in which the neuron's firing exceeded the threshold, while FPR reflects the proportion of "cat-like" trials in which firing also exceeded that same threshold. The AUC was calculated as the integral of the ROC curve and serves as a measure of how well the neuron discriminated between the two stimulus categories based on firing rate alone.

Other places where there is a lack of clarity:

- Figure 2C: the figure and caption are very unclear. Where are the image onset, red square onset, and possible response window onset in the timeline? Why does the monkey appear to be releasing the bar at approximately the same time for all but the most doglike of images? Why aren't the bar release times delayed, as the morph level increases (and images are more catlike)? Why does it appear that even for the cat-like images (morph levels under 50%), monkeys were releasing the bar in the same interval as for the dog-like images?

We thank the reviewer for pointing out the lack of clarity in Figure 2C and its caption. In the revised figure, we have added the onset of the red square, which marks the beginning of the response window. Time 0 corresponds to the image onset. This panel focuses on events occurring early in trials, to highlight the time window relevant to the neural analyses presented in this study. Events in the 2nd interval are not shown.

For the most dog-like images, monkeys typically waited and responded in the second interval (not shown in this figure). In contrast, for ambiguous dog-like images near the 50% morph level, monkeys were more likely to make errors by responding in the first interval, which overlaps with correct responses to cat-like images.

As for cat-like images, when the morph level increases and the images become more ambiguous, we observed a modest increase in reaction time (see Fig. S1B). This is also reflected in Figure 2C, where there is a slight delay in bar release timing for more ambiguous cat images compared to the most cat-like ones.

In the Figure 2C legend on line 871 - 876, we added: Right panel: spike raster (black dots), onset time of red square (red dots) and bar-release time in the first interval (blue dots) when images in 10th morph identity were presented. Time 0 corresponds to image onset. Events in the second interval are not shown

- Figure S1B, are the RTs for the dog-like images measured with respect to the start of the SECOND interval? (this is not stated).

Yes, we have added the definition in the figure legend on line 1005 - 1007: Reaction times for correct bar releases in the second interval for dog-like images were defined as the time between the onset of the green square and the moment the bar was released.

- Figure 2E – the calculation of the “Effect size of visual category” needs better explaining

We have added explanation in the figure legend on line 883 - 884: Omega-squared (w^2) effect size of visual category is an unbiased estimator of the proportion of variance in the neural activity that is attributable to visual category (see Methods).

We also rephrased the calculation of the “Effect size of visual category” in methods to make it clearer on line 559 – 568: To quantify how strongly neural responses were modulated by category, we computed the omega-squared (w^2) effect size based on outputs from a one-way ANOVA. Omega-squared is an unbiased estimator of the proportion of variance in neural activity that is attributable to a specific factor (e.g., visual category), while accounting for degrees of freedom and error variance. It is calculated as:

$$w^2 = \frac{SSQ_{effect} - df_{effect}MS_{error}}{SSQ_{total} + MS_{error}}$$

SSQ_{effect} represents the sum of squares of the factor of interest (e. g. category), df_{effect} is the degrees of freedom associated with that factor, MS_{error} is the mean squared error and SSQ_{total} is the sum of squares across all factors. Omega-squared values range from –1 to 1, with higher positive values indicating stronger modulation of neural responses.

- A better definition (appearing earlier) is needed for “morph identity”. It took a long time

for me to realize that this refers to one specific morph sequence with a unique pair of dog-cat parents. Clarify earlier.

Thanks for the suggestion. We now clarified the definition where morph identity was first mentioned on line 59: Morph identity refers to one specific morph sequence with a unique pair of dog-cat prototypes

- Figures 3 and 4: clarify explicitly whether asterisks indicate test of significance between TE vs TEO or between day 1 and day 9 within a given brain region

We now clarify explicitly in Figure 3 legends (Figure 5 in the revised version) on line 978

- 982: (E) and (F)..... Magenta asterisks alongside the TE curve indicate a significant correlation between dog AUCs and day number for TE. Red asterisks next to the brace spanning the two curves indicate significant interactions between brain region and day number. Black-over-magenta and magenta-over-black asterisks atop the curves indicate $TEO > TE$ and $TE > TEO$, respectively.

Figure 4 (Figure 3 in the revised version) on line 902 - 909: (B)..... Magenta asterisks alongside the TE curve indicate a significant correlation between TE decoding accuracy and training day.Red asterisks next to the brace spanning the two curves indicate significant interactions between brain region and day number. Notations are consistent across all related figures.

- Some of the methods need editing by a native English speaker – grammatical errors

We have carefully revised the manuscript for grammatical accuracy and clarity and have had the text reviewed and edited by a native English speaker to improve overall readability.

- Lines 328 to 330: text implies that the optimal response strategy was simply to always release the bar as soon as the square turned green (never during the red square). This is obviously not quite correct. It depended whether the image was cat-like or dog-like (if cat-like it makes sense to release early to move on to the next trial, but if dog-like it's optimal to wait). Explain this here so that the reader is not confused.

We explain the optimal strategy on line 511-516: The optimal response strategy for monkey X depended on the visual stimulus, which was the same as for monkey T. For cat-like images (which were never rewarded), releasing early allowed the monkey to efficiently skip the non-rewarding trial. In contrast, for dog-like images (which could lead to reward), it was advantageous to wait until the green square (second interval) before releasing the bar to obtain a reward.

- Lines 420 – 421 unclear which images were being used. Please edit for clarity.

The traditional decoding strategy refers to Figure 3A. we now clarified it on line 626 in the revised manuscript.

Reviewer #3 (Remarks to the Author):

This study addresses an interesting question and was exciting to read. It is well written, with good design elements and comprehensive set of analyses, and several of the findings are compelling. The data show that TE category representations increase over the course of learning, tracking performance both over time and on trial-by-trial basis. My main concerns regard robustness, lack of nuance in treating the results, and potential limitations of conclusions driven by the experimental design. I would also like to get a better idea of the authors' thoughts about the significance of this TE/TEO results on a mechanistic level. Let me clarify those in more detail.

Thanks for the reviewer's insightful comments. We respond to each concern point-by-point in the text below.

1. Robustness. Some of the key conclusions hinge on results that are based on as few as 8 TE neurons in Monkey T. The noisiness of that data is apparent in several figures. Some of the key conclusions hinge on days 7-9 in Monkey T, and would have been different if Monkey T would only have 5 days of recording like Monkey M had. Some of results can only be seen for dog but not cat images, even in monkey where cat and dog categories are supposed to be balanced. All of this makes me less confident in the robustness of the results.

I understand the challenges of obtaining data like this, but more could be done to acknowledge those limitations (beyond the asymmetry for cats/dogs discussed currently). My concerns are eased somewhat by the population coding analyses that I see helping somewhat with the small-N TE neuron count in Monkey T. The authors could consider starting their results with those, to establish the main pattern of findings, and then provide the deeper dive focusing on single cell-based analyses that are interesting and informative, but need to a little more careful framing.

Thanks for the reviewer's suggestions. We now reconstruct the manuscript, moving the single cell-based analyses after population results.

2. Lack of nuance in interpreting results. Eventually, as a whole, I mostly accepted the key conclusion of the paper, especially as it drew from the performance-based analyses (link to behavior) and changes over time. The abstract does a good job making clear what key findings are drawn upon for conclusion. However, conclusions of several individual analyses don't seem justified and the data are painting a potentially interesting picture that is never mentioned. Examples:

- Lines 103-110 note that category selectivity was similar between TE and TEO, but TEO had greater mixed selectivity, so that means that TE is the category selective one. That's not necessarily an obvious conclusion.

Thanks for the reviewer's concern. We now rephrase the interpretation more cautiously on line 230-234: These results suggest that relative to TEO, neurons in TE were more likely to exhibit responses dominated by a single category. Although both areas encode category information, the reduced mixed selectivity in TE may point to more specialized category representation compared to the more multiplexed coding in TEO.

- Figure 3 related analyses emphasize that category separation increases over training in TE. But a closer look suggests that category separation already starts high in TEO, so TE is just catching up in category representation that is already starting strong in TEO. Then, the results are not holding up for cat in Monkey T (where cat/dog are both rewarded). And in Monkey M, TEO does show greater AUC (category separation) over TE even for cat, across all days. The conclusions don't really address all these

nuances, or the fact that TEO is showing strong category separation early. Given prior lesion findings that both TE and TEO contribute to category learning and potentially interact (largest effect when both are lesioned) could be maybe linked to this finding of early category representations in TEO. Even if it's not currently clear how the regions may interact, this piece of data may prove valuable to our understanding eventually.

Thanks for the reviewer's insightful observation and thought. We now did statistic tests on AUC results between TE and TEO for each day. The results were not consistent between two monkeys (Figure 5E,F, Figure S8H,I), which may be caused by the limited neuron number in monkey T. However, we did observe consistent results across two monkeys on the population decoding that category separation starts high in TEO, TE is just 'catching up' in category representation (Figure 3B, 3D, inset for monkey X; Figure 3E, left panels. For consistency with our previous publication (Pearl JE, *et al.*, 2014) and to facilitate comparison, 'monkey M' has been renamed 'monkey X' in the revised manuscript).

We have added these results on line 140 - 144: Notably, category representation was initially stronger in TEO than in TE (Figure 3B, 3D; inset for monkey X; Figure 3E, left panels). As learning progressed, representation in TE surpassed that in TEO. This temporal crossover suggests that TEO may support early, coarse category representation, while TE undergoes experience-dependent plasticity to refine the encoding of well-learned categories (see Discussion).

Also, related discussion was added on line 323 – 341: Mechanistically, we interpret our results as evidence that plasticity in TE supports the formation of more abstract, experience-dependent, behaviorally related category representations. In contrast, TEO may support simpler visual feature processing, which is less plastic, and is hence less influenced by learning.

Based on our findings—that category-related signals in TEO are stronger than those in TE during the early stage of learning (Figure 3B, D, E), that TE alone shows a gradual, learning-related increase in decoding accuracy, we propose that TE and TEO play complementary roles in category learning. Representation of category in both TE and TEO is significantly stronger than chance level (Figure 3B, D). In both monkeys, and across decoding strategies, the category

discriminability of pooled TE and TEO neurons—i.e., $d'(TE+TEO)$ —often exceeded that of the stronger individual region (Figure S9, Figure S10E, N, Figure S11D, H). Specifically, at the initial stages of learning, TEO may provide an early, coarse category representation sufficient for basic categorization demands, perhaps based on simple visual features. As learning progresses, TEO, as well as V4 which directly projects to TE¹⁴, continues to supply lower-level visual feature representations, which are refined and integrated within TE to gradually construct more abstract and behaviorally relevant category representations. This interpretation is consistent with prior lesion studies showing that bilateral removal of either TE or TEO causes only mild deficits, while combined lesions produce severe, long-lasting impairments, highlighting the interdependent roles of these two regions.

- Similar issue/nuance exists in Category decoder analysis, where category decoding in both monkeys starts stronger in TEO than TE. See also my comments on design where the early TEO category representation could come from. While the abstract is careful in verbalizing the findings, the title may be a little overreaching given the existence of category representations in TEO that may precede category representation in TE. I understand the argument of behavioral relevance but we cannot rule out TEO contributions to category learning given the design and data.

Thanks for reviewer's comments. We now rephrase the title as '**Enhanced Neural Plasticity in TE Compared to TEO During Learning of a Feature-Ambiguous Visual Categorization Task**'. We also discussed the effects of design (extensive pre-training) on the early TEO category representation in the following responses.

3. Design. There are a couple of aspects of the design that need to be emphasized more prominently than they currently are: the different procedures for the two monkeys and the pretraining regimen.

- Different procedures. I understand that the difference was not intended to be a manipulation in its own regard. Nevertheless, the difference between one-category learning (A/notA) and two-category discrimination (A/B) is an area of its own investigation, as least in human primate literature. While the tasks could be in principle treated the same, empirical research in humans suggests otherwise. For example, A/notA can be learned by amnesiacs and recruits striatum and perceptual learning

system more heavily while A/B cannot be learned by amnesiacs and recruits hippocampus more heavily. Thus, for broader impact on the field, it's important that this paper makes it more apparent that both of these category learning tasks were used, A/B in one monkey and A/notA in another. While I understand this may not make it into the abstract, a more explicit notion in the text (plus a link to the human literature that dissociates them) is needed. Figure 1A should be also edited to make it clear the design only applies to one Monkey and what was the difference for the other monkey. Rationale for this difference should also be offered: was it intentional? Accidental? It is exciting to see the results that were consistent across monkeys. It may also be fruitful in the future to consider some of the inconsistent findings in light of these procedural differences, even though they are currently not attributable to the procedure difference based on a single monkey undergoing each procedure.

Figure 1A was edited to include both procedures.

We discussed on line 431 – 442: Different training procedures may have encouraged the two monkeys to adopt different categorization strategies: an A/not-A strategy (only one category rewarded) for monkey X, and an A/B strategy (both categories rewarded) for monkey T. While we cannot directly confirm the strategies monkeys used, the reward structure likely biased the animals toward these approaches. In human studies, A/not-A and A/B categorization tasks are known to recruit distinct learning systems: A/not-A tasks tend to rely more on the striatum and perceptual learning systems, whereas A/B tasks more strongly engage the hippocampus and declarative memory systems^{39,40}. However, our findings reveal consistent TE-related plasticity during learning across two monkeys, suggesting that in non-human primates, visual categorization involving ambiguous stimuli and fine-grained discrimination may ultimately recruit overlapping neural processes in the IT cortex—regardless of the learning strategy. This possibility warrants further investigation in future studies.

We added one paragraph in method section on line 517 – 521: The use of different procedures across monkeys was not an intentional manipulation aimed at testing the distinction between A/notA and A/B, but rather a result of the progressive development of our training protocols. Monkey X was trained earlier using a more traditional A/notA design. For monkey T, we refined the procedure based on our prior experience and implemented a more balanced A/B design to reduce potential category biases.

- Extensive pre-training. As the authors noted, this study would not be possible without extensive pretraining as trainings can take weeks to months. However, it is important to mention the extensive pre-training prior to methods, and consider it during interpretation/discussion of the results. We do not know how these regions contributed to that initial learning, so it may have an important implication for the findings. It's maybe a design aspect responsible for the strong category representations existing in TEO already on day 1, which may require further nuance in the interpretation/discussion.

We thank the reviewer for this important and thoughtful comment. We now added discussion on line 342 – 353: It is important to consider the potential impact of extensive pre-training on the initially stronger category representation observed in TEO during the first day of the morphed cat/dog categorization task. Prior to the morphed cat/dog categorization task, monkeys underwent months of training to become familiar with the task structure, fixation

requirements, and general categorization demands. They were trained on non-morphed cat/dog categorization tasks before progressing to the morphed versions. For monkey X, we recorded data during the first-ever cat/dog categorization task (20-cat/20-dog, Figure S10). Even in this early stage, decoding accuracy was consistently higher in TEO than in TE, suggesting that the strong initial category representation in TEO is not solely attributable to extensive pre-training on cat/dog tasks. However, we cannot entirely rule out the influence of other pre-training experiences, such as fixation training or earlier exposure to Walsh pattern categorization tasks, which may have contributed to the development of initial category-selective responses in TEO.

4. Big picture. The general discussion was somewhat limited in scope, discussing one limitation (there are more not mentioned) and not offering much in terms of significance, or how the circuitry may work mechanistically. Can any more thoughts be offered? Why TE rather than TEO when we focus on changes over time and link to behavior?

Thanks for the reviewer's insightful suggestions. We now added mechanistical discussion on line 323 – 341 (presented in responses to the previous concerns), as well as discussion on the greater plasticity of TE on line 354 – 364: The greater plasticity of TE compared to TEO during learning may stem from its anatomical position and broader connectivity. TE neurons have larger receptive fields and integrate bottom-up inputs from simple visual features, allowing them to respond to complex stimuli and support more abstract, flexible representations^{14, 34}. In contrast, TEO neurons have smaller receptive fields and respond to simpler features, providing stable encoding of low- to mid-level visual features^{14, 35}. Moreover, TE exhibits broader connectivity than TEO, including reciprocal connections with the medial temporal lobe, orbitofrontal cortex, and ventrolateral prefrontal cortex—regions involved in memory, reward processing, and decision-making¹⁴. These widespread connections may facilitate top-down modulation in response to behavioral relevance, enabling TE to adapt its representations through learning. In contrast, the more limited connectivity of TEO may constrain its capacity for such learning-dependent plasticity.

5. Smaller comments

- Figures were generally of high quality. Figure 1 should have the two different

procedures visualized or somehow noted, given my prior comment. Figure 7S may be more useful is some other visualization (maybe line graph) given that displays decoding accuracy with chance of 50%. The bargraph starting at zero makes it difficult to see any deviations from chance, and the line graph still allows visualization of differences between conditions.

Figure 1 was edited to include two different procedures. Figure S7 (Figure S6 in revised version) was now updated with line graph.

- In general, it would be helpful to provide more rationale for some of the design choices and some explanations.

We added the rationale in the method on line 517 – 521 (presented in response to previous concerns)

- Why were there so few TE neurons sortable from two arrays in Monkey T, while substantially more were captured by one array in TEO in that same monkey, or even more by the arrays in the other monkey?

We thank the reviewer for the question. Although the implantation procedures and hardware (chronically implanted Utah arrays) were consistent across animals and cortical regions, chronic array performance can vary considerably due to several factors, including positioning during surgery, local cortical anatomy, vascularization, and long-term tissue responses such as gliosis or scarring. In the case of Monkey T, the lower yield from the two TE arrays may reflect less optimal array placement, suboptimal electrode-tissue interface, or reduced recording stability over time.

Reviewer #4 (Remarks to the Author):

In this manuscript, the authors use chronic recording arrays in two areas of macaque temporal cortex (TE and TEO) in combination with relatively rapid category learning, to try to establish the nature and distribution of signals that may support the establishment of new categories. The question is very significant: while there are many theories and data about category tuning (including some human fMRI studies), there is far less data on the emergence of categories through learning, and in particular whether there are differences between TE and TEO. Recent lesion studies, cited in the introduction, point to a parallel processing of signals supporting categorization between the two areas, and the authors wanted to examine this during the actual process of learning, in which the complications of interpreting lesions data (such as targeting and compensatory mechanisms) are eliminated. The authors have an impressive data set, appropriate analyses at both the single cell and ensemble level and some nice behavioral evidence of learning, but inconsistencies between the animals, as well as imprecise language with regard to categorization computation (as opposed to attribute encoding), detract from its potential impact.

Specific Comments:

1. The task design, with its temporal structure, certainly makes for numerous potential confounds, including pre-motor, reward anticipation, and passage of time encoding, and by in large the authors do a good job of trying to address many of these in the discussion. But there were training differences between the animals in terms of the task, and also several instances, where the animals' physiology was not consistent, and a concise discussion of whether the two (training and physiology differences) could be

related is warranted. especially since there are numerous differences between animals. Examples include morph identity neurons (Fig S3) , category selectivity (Fig S5B&E), and error effects (Fig S7B). These differences can have major implications: looking at S7 for example, it is clear that TEO “conclusion” (that it doesn't have performance related information) relies on a single animal. In general, given all these differences, the authors need to be much more circumspect in their statements. For example, specifically with regard to the error analysis, it is concluded that TEO is not a “plausible candidate region for the computation of category membership.” How does Monkey M feel about that?!

We thank for the reviewer's comments. We now rephrase the conclusion in the error analysis section on line 211 - 215: The enhanced category information in the TE population in correct trials versus error trials indicates that TE is a plausible candidate region for supporting downstream computation of category membership of the stimuli used in this study. Further investigation is needed to clarify the contributions of TEO given individual variability across animals.

We now added discussion about the inconsistent aspects between two monkeys on line 443 – 451: We observed several inter-animal differences including: neurons modulated by morph level/identity (Figure S3), category selectivity neurons (Figure S8B&E), dog generalization (Figure 3E), and error effects for TEO (Fig S6B&D). These differences may stem from differences in task design (e.g., reward contingencies), species differences, and/or the unequal number of recorded neurons - substantially fewer neurons were recorded from monkey T, which limits our statistical power to detect learning-related changes. Despite these individual differences, our key finding — enhanced neural plasticity in TE compared to TEO during learning of our ambiguous visual categorization task — was consistent across both subjects, demonstrating that the core effects generalize across animals despite individual variability.

2. Speaking of computing categories, I think the authors are on dangerous ground with such statements. As demonstrated by Kobatake and Sigala, attribute tuning is subject to learning, and with population decoders you can get category performance even from signals selected solely according to their shape preferences (Bougou et al, 2024). So if a region's attribute tuning changes that helps categorization, but does not imply that

categorization is computed in that region: any task in which that attribute is relevant (discrimination, detection in noise) would also be helped. Looking at the few examples in individual cells that are presented (more on that later), I really don't see anything that I would consider a category cell (that fires the same across morph identity and only varies in morph strength).

We thank the reviewer for highlighting this important point.

We now removed the phrase of category computation and updated it on line 211 – 215: The enhanced category information in the TE population in correct trials versus error trials indicates that TE is a plausible candidate region for supporting downstream computation of category membership of the stimuli used in this study. Further investigation is needed to clarify the contributions of TEO given individual variability across animals.

We also added discussion on line 365 – 374: The observed enhanced category decoding accuracy in TE, along with an increase in TE population similarity for stimuli within the same category, suggest that learning sculpts TE representations to better reflect categorical structure. One possible mechanism underlying this reorganization is enhanced encoding of shared visual features among category members. This aligns with prior findings showing that attribute tuning is shaped by learning^{27, 28} and that population-level category decoding can arise even from neurons primarily tuned to simple shape features, without requiring explicit categorical coding³⁶. Thus, while such reorganization supports category-based discrimination, it does not necessarily imply that TE computes category membership per se. Instead, TE may provide structured sensory representations that facilitate downstream decision-making processes.

3. In this regard, it would be nice to look at, in both individual neurons and a population level, trial-by-trial choice effects for the 50% (even near 50%) morphs. While positive choice probability is challenging to interpret (attention vs read-out), the lack of any choice signals would seem to constrain how strongly these signals contribute to the actual categorical choices made by the animals.

We thank the reviewer for this insightful suggestion. We analyzed trial-by-trial choice-related signals during ambiguous trials (50% morphs), both at the single-neuron and population levels.

At the single-neuron level, we computed choice probability using ROC analysis. The proportion of neurons with significant choice probability in either TE or TEO remained around chance level. This result is consistent with previous findings (Mogami & Tanaka, 2006), suggesting that single-neuron activity alone may not reliably reflect categorical choices for ambiguous stimuli.

At the population level, we observed a different pattern. Significant choice-related decoding accuracy was found on multiple days in both monkeys, particularly in TE (TE: 5 out of 9 days, all 5 days for monkey T and X, respectively; TEO: 2 out of 9 days, 4 out of 5 days for monkey T and X, respectively. p values were adjusted by Benjamini-Hochberg FDR correction. For consistency with our previous publication (Pearl JE, et al., 2014) and to facilitate comparison, 'monkey M' has been renamed 'monkey X' in the revised manuscript). These results suggest that, although individual neurons do not consistently encode categorical choices during ambiguous trials, population-level activity in IT cortex relatively reliably reflects the animal's decisions.

Figure S5

We now add method for choice probability on single neurons and choice decoding on populations on line 672 – 686:

Choice probability and Choice decoding

Choice probability was computed for each neuron using ROC analysis. Trials were grouped based on the animal's choice (i.e., bar release in the first vs. second interval), and for each neuron, we calculated the area under the ROC curve (AUC) to quantify how well firing rates discriminated between choices. The second-interval choice was designated as the positive class. To enable direct comparison of choice selectivity across neurons, AUC values < 0.5 (favoring first-interval choices) were transformed to $1 - \text{AUC}$, such that all AUCs > 0.5 reflected choice discrimination strength regardless of direction. We assessed significance using permutation tests by shuffling choice labels across trials.

To train the choice decoder (using the same procedure described in the “*Category decoding in the neural population*” section), trials were grouped according to the animal's choice, regardless of image identity. For each choice condition, trials were randomly divided into training (80%), validation (10%), and test (10%) sets. To assess chance-level performance, choice labels in the training set were randomly shuffled. This decoding process was repeated 500 times for each learning day.

We added one section in the results on line 169 – 189:

Choice-related signals in TE and TEO

We further quantified how well the responses of both single neurons and neuronal populations predicted the animal's category choice on a trial-by-trial basis (see Methods). We focused our choice probability analysis on 50% morph (half dog and half cat) trials, which allowed us to dissociate neural activity related to internal choices from categorical stimulus-driven responses. The proportion of neurons with significant choice encoding in either TE or TEO remained at chance level (Figure S5A). This result is consistent with previous findings³², suggesting that single-neuron activity in IT may not reliably predict category choices for ambiguous stimuli.

At the population level, we observed a different pattern. Significant choice-related decoding was observed on multiple days: for monkey T, TE and TEO on 5/9 and 2/9 days, respectively; for monkey X, TE and TEO on all 5 days and 4/5 days, respectively (Figure S5B, p values were adjusted by Benjamini-Hochberg false discovery rate (FDR) correction). The proportion of days in which category choice could be predicted from the neural responses was not statistically different between TE and TEO for either monkey (Fisher's exact test, $p > 0.05$ for both monkeys). These results suggest that, although individual neurons do not seem to encode categorical choices during ambiguous trials, population-level activity in IT cortex relatively reliably reflects the animal's decisions.

Interestingly, although category representation increased over training (Figure 3B, D), choice signals at the population level were not shaped by learning (Figure S5B). These results imply a functional dissociation between category representation and choice-related activity in IT cortex during learning.

We add discussion on line 375 – 380: Although category decoding accuracy in TE increased significantly over learning and correlated with behavioral performance, choice decoding remained stable across training days. This dissociation suggests that learning selectively enhances the representation of category structure in TE, reflecting experience-dependent plasticity. In contrast, trial-by-trial choice signals may form in downstream decision-related areas. These findings highlight distinct neural mechanisms underlying categorical learning versus moment-to-moment decision processes in the IT cortex.

4. The authors do a good job of testing the decoders within and across categories with the dissimilarity metric (Fig 5E). But what about consistency with Fig S1A: shouldn't the decoders be much worse at morphs near 50% than they are ones that aren't?

Yes. The proportion identified as dog by the decoder at each morph level was plotted in Figure S4. Decoders were much worse at morphs near 50% than they are ones that aren't, except for TEO of monkey T, for which the tendency was less obvious.

Figure S4

5. The authors have a lot of data, but in some cases only examples are shown where I

think a summary could be shown as well. For example, with regard to morph identity/level tuning (Fig 2C), why not, as a function of training (or even first day vs last day), show a population summary where morph identity and morph level are sorted for each neuron, normalized to the peak response, and averaged? It would be nice to know if morph identity tuning is becoming broader and morph level tuning more binary (both consistent with categorization) with learning.

We plot the mean normalized (to the peak response) responses across populations without and with morph identity and morph level sorted (first sorted by morph identity, then sorted by morph level separately for cat and dog).

It is difficult to identify changes in the TE plot with morph identity and morph level sorted for monkey X. But for both monkeys, the changes in TE populations are obvious for plots without morph identity and morph level sorted. Thus, we show the version without morph identity and morph level sorted in Figure 2D.

We observed an overall increase in responses to dog images for both monkeys, which is consistent with the ROC analysis in Figure 5.

We add description on this figure in the results on line 88 - 91: **Mean normalized responses across the TE neuron population, but not that of TEO, showed greater separation between responses to dog-like and cat-like images on the last day compared to the first (Figure 2D). This increased separation was primarily driven by an overall enhancement of responses to dog-like images, as detailed in the following sections.**

I'm also not sure where the population decoding neuronal sample size in Fig 4 comes from: are the 8 and 59 neurons those neurons which are thought to be present in every single day's recordings? How was neuronal identity across days verified? And what is the population in TE and TEO respectively?

The neuronal populations recorded on each day were not identical, as neural identity could not be definitively tracked across days due to the limitation of chronically implanted Utah arrays. The number of neurons recorded from TE and TEO on each day varied (see Figure 2B for detailed counts). To enable fair comparisons of population decoding performance across days and brain regions, we employed a subsampling approach: we randomly selected the same number of neurons across all conditions based on the minimum number of simultaneously recorded neurons in any condition. This resulted in fixed sample sizes for decoding (8 neurons for Monkey T and 59 neurons for Monkey X).

What happens if I just through every neuron recorded in the few days and compare that with every neurons recorded in the last few days? And is there any chance that decoding differences between the animals (for example, the lack of within dog similarity in Monkey T Fig5E) are simply constrained by smaller sample sizes?

We thank the reviewer for this thoughtful suggestion. We pooled neurons recorded from early (days 1 - 4) and late learning days (days 6 - 9) in Monkey T and recomputed representational dissimilarity for each period. The results remained consistent with our original findings.

As the recordings were obtained using chronically implanted Utah arrays, the neurons sampled across days likely came from overlapping or nearby cortical locations. This suggests that their functional properties remained relatively similar across days, and that pooling across days is unlikely to introduce new information. The absence of learning-related changes in representational dissimilarity for dog stimuli in Monkey T's TE is likely due to limitations in neural sampling—possibly due to sparse or biased coverage—which may fail to capture the true population-level representational dynamics.

6. The manuscript starts with the conclusion based on lesion studies that there is a “parallel” system for categorization between TE and TEO, but this point seems inconsistent with how the discussion and abstract are framed, which strongly emphasize TE. But could the parallel scheme be tested by computing d' for TE and TEO ensembles separately, and then together, and seeing if the together d' is consistent with independent detectors?

We thank the reviewer for this insightful suggestion. We calculated d' based on scores from the decoder. We observed that in both monkeys and across both decoding strategies, $d'(\text{TE}+\text{TEO})$ exceeded the stronger individual region on several days. On the remaining days, $d'(\text{TE}+\text{TEO})$ was either equal to or less than the stronger region alone.

These results indicate that TE and TEO can contribute to categorization in a complementary manner, but this benefit is not consistently expressed—potentially due to day-to-day variability in neural signals, limited neuronal sampling (particularly in Monkey T).

Figure S9

We add methods on line 723 – 738:

Quantifying discriminability of neural ensembles with d'

To quantify category discriminability of neural population responses, we computed the discriminability index (d') from decoder output scores for TE and TEO ensembles individually and in combination. For each decoding repeat (200 repeats in total), d' was calculated as follows:

$$d' = \frac{|\mu_{cat} - \mu_{dog}|}{\sqrt{0.5(\sigma_{cat}^2 + \sigma_{dog}^2)}}$$

μ_{cat} and μ_{dog} are the means of the decoder output scores for test trials from two different categories (e.g., cat vs dog). σ_{cat}^2 and σ_{dog}^2 are the variances of those scores within each category.

For individual regions, decoding followed the same procedure as described above. For combined analyses, we randomly selected and pooled the same number of neurons from TE and TEO as used in individual decoding (Monkey T: 8 neurons from each region; Monkey X: 59 neurons from each region). Interpretation of d' followed the following logic: $d'(\text{TE+TEO}) \approx \max(d'(\text{TE}), d'(\text{TEO}))$ suggests redundant or overlapping information. $d'(\text{TE+TEO}) > \max(d'(\text{TE}), d'(\text{TEO}))$ indicates complementary, partially independent contributions. $d'(\text{TE+TEO}) \approx d'(\text{TE}) + d'(\text{TEO})$ reflects fully independent, non-overlapping information. $d'(\text{TE+TEO}) < \max(d'(\text{TE}), d'(\text{TEO}))$ implies that one region may introduce noise or task-irrelevant signals that reduce discriminability.

We add results on line 250 – 266:

TE and TEO can contribute to category discriminability in a complementary fashion

Our lesion studies suggest that TE and TEO may function in parallel during categorization. To evaluate this, we computed a discriminability index (d') from decoder scores for TE and TEO ensembles individually and in combination. Compared to decoding accuracy, d' quantifies the separation between category distributions relative to their variability, offering a more sensitive and non-saturating measure of neural discriminability. Each region's d' profile mirrored its decoding-accuracy pattern (Figure S9A, B vs Figure 3B, D, $p_{(\text{matrix} \times \text{day})} > 0.05$ for all comparisons, LM). In both monkeys, and under both decoding strategies, $d'(\text{TE+TEO})$ exceeded the stronger individual region - $d'(\text{TE})$ or $d'(\text{TEO})$ - on several days (Monkey T: 3/9 and 3/9 days; Monkey X: 4/5 and 2/5 days for category and generalization decoders, respectively), yet remained below the sum of the two regions, i.e., $d'(\text{TE}) + d'(\text{TEO})$ (Figure S9). This pattern

indicates that TE and TEO provide complementary, though partially overlapping, information that can enhance category representation when combined. On the remaining days, d' (TE+TEO) was either equal to or less than the stronger region. These results indicate that TE and TEO can contribute to categorization in a complementary manner, but this benefit is not consistently expressed—potentially due to day-to-day variability in neural signals, and/or limited neuronal sampling (particularly in monkey T).

We rephrase the abstract on line 25 - 29:Further, TE and TEO can contribute complementary, partially independent category information. Combined with the findings from lesion studies, our results suggest that although TE and TEO can represent complementary information, neural plasticity in area TE likely plays a more fundamental role in supporting visual category learning.

We also discuss it on line 329 – 341:

.....we propose that TE and TEO play complementary roles in category learning. Representation of category in both TE and TEO is significantly stronger than chance level (Figure 3B, D). In both monkeys, and across decoding strategies, the category discriminability of pooled TE and TEO neurons—i.e., d' (TE+TEO)—often exceeded that of the stronger individual region (Figure S9, Figure S10E, N, Figure S11D, H). Specifically, at the initial stages of learning, TEO may provide an early, coarse category representation sufficient for basic categorization demands, perhaps based on simple visual features. As learning progresses, TEO, as well as V4 which directly projects to TE¹⁴, continues to supply lower-level visual feature representations, which are refined and integrated within TE to gradually construct more abstract and behaviorally relevant category representations. This interpretation is consistent with prior lesion studies showing that bilateral removal of either TE or TEO causes only mild deficits, while combined lesions produce severe, long-lasting impairments, highlighting the interdependent roles of these two regions.

7. TE has been divided in to TEpd (dorsal posterior) and TEad (dorsal anterior) on the basis on connectivity (for example Saleem et al, 2000, Kravitz et al, 2013) and it looks the authors' have arrays in both. Any differences?

In our study, array placements spanned both anterior and posterior portions of TE. However, for monkey X, the anterior array yielded too few well-isolated units to support robust statistical analysis.

Among the remaining two arrays, we compared the middle and posterior sites and found that the middle array exhibited stronger generalization category decoding, suggesting a potential anterior-posterior gradient in category abstraction within TE.

For Monkey T, the number of recorded neurons was generally limited, and neuron counts from both the anterior and posterior arrays were insufficient to allow meaningful subregional comparisons. We agree that investigating functional distinctions across TE

subregions is an intriguing direction and warrants further exploration with larger-scale recordings in future studies.

We thank the reviewers for their positive evaluations. We appreciate the constructive feedback throughout the review process, which helped improve the quality and clarity of the manuscript.

Reviewer #1 (Remarks to the Author):

the authors have adequately answered my comments, and those of the other reviewers; the collection of novel data is especially welcome

We thank the reviewer for the positive evaluation and are especially grateful for the recognition of our efforts in collecting novel data.

Reviewer #2 (Remarks to the Author):

In general, the authors have done a good job addressing my concerns. The new analyses examining the interaction of brain region by day have greatly improved the manuscript and in most cases the authors' claims of enhanced learning are now justified. There have also been many clarifications which improve the manuscript's readability. I have just a few remaining minor concerns.

On at least two occasions, the "brain region x day" interaction was significant in one animal but not the other. This is somewhat glossed over, as though the result were found in both animals:

- On page 4: "In monkey T, the proportion-versus-day correlation was significantly stronger in TE than in TEO, whereas in monkey X the same trend did not reach significance" but later in the paragraph "Thus, it appears that category-modulated neural activity in TE, but not TEO, increased during learning"
- On page 9: "which indicated that encoding of one category (dog images) was enhanced in TE during learning"  only true in one of two animals.
- In both cases, the authors need to state in the final conclusion, not just buried in the results part, that it was only found in 1 of 2 animals.
- Needs to be mentioned again on page 15 where inter-animal differences are listed.

We appreciate the reviewer pointing out this inconsistency.

On page 4, line 104 – 106, we update the conclusion: Taken together, these analyses show that TE's category signal strengthened with learning in both monkeys: the TE-versus-TEO difference is robust by effect-size in both monkeys and additionally confirmed by the percentage-of-significant-neurons metric in monkey T.

On line 280 – 282, we update the conclusion: This TE-specific growth was significant in monkey T and showed a similar but non-significant trend in monkey X, indicating that the enhancement of dog-image encoding was robust in one animal and partial in the other.

On line 462 – 464, we mentioned these inter-animal differences: proportion of neurons modulated by category (Figure 2E), dog-image encoding (Figure 5E&F).....

Relatedly, on page 16, where it is stated “our key finding — enhanced neural plasticity in TE compared to TEO during learning of our ambiguous visual categorization task — was consistent across both subjects” needs to be tempered/watered down. It was not *consistently* found because in at least two cases, the effect of *learning* was found in only 1 of 2 animals. I appreciate that it was “often” found, but I think the wording needs to be made a little more circumspect. Perhaps “typically found”, or “found in most analyses” or similar?

We update the statement on line 468 – 471: ...our key finding — enhanced neural plasticity in TE compared to TEO during learning of our ambiguous visual categorization task — **was consistent in most analyses** across both subjects...

Figure 1A (the experimental task) is very unclear. It looks like Monkey X (denoted by dashed lines) only has the option to do the same thing for both cat-like and dog-like, and in either case the only outcome is to go to the next trial. This cannot be right. It contradicts the description in the methods: “The optimal response strategy for monkey X depended on the visual stimulus... For cat-like images (which were never rewarded), releasing early allowed the monkey to efficiently skip the non-rewarding trial. In contrast, for dog-like images (which could lead to reward), it was advantageous to wait until the green square (second interval) before releasing the bar to obtain a reward.” I suggest revising this figure.

We thank the reviewer pointing out the unclarity in the figure. We now update it.

A

Line 576: "increasement" (not an English word)  "increase"

Thank you for catching that typo. We have corrected "increasement" to "increase".

Overall, this work will now make a valuable contribution to the literature.

We thank the reviewer for the kind words and are pleased that the work is now seen as a meaningful contribution to the field.

Reviewer #3 (Remarks to the Author):

The authors did a good job addressing many of my and other reviewers' comments. While more data would have been ideal, I recognize that science is incremental and cumulative, and the current paper makes a sufficient contribution within the scope of the data available.

We thank the reviewer for their thoughtful feedback and appreciation of our efforts. We're pleased that the contribution is seen as meaningful within the scope of the available data.

I just suggest to acknowledge the category signals in TEO (pre-existing those in TE) in the abstract, in addition to acknowledging them in text. As discussed, all the TEO effects could have happen during pre-training and not be observed here for that reason. The abstract, as written, may provide a somewhat misleading impression regarding the role of TEO.

We now update the abstract on line 19 – 20:**Category specificity and generalization were initially stronger in TEO than in TE but increased with learning only in the TE neuronal population.....**

Reviewer #4 (Remarks to the Author):

I appreciate and applaud the rigor with which the authors have added additional analyses and modified their conclusions, but, especially with consideration of these new additions, do not find the central tenet, as specified in the final line of the abstract, that TE plays a “more fundamental role” in category learning, well justified by the data. Indeed, the change in the title, which now talks about plasticity during learning, rather than explaining learning, makes me wonder if the authors still feel strongly about this claim given their data. This is a complicated data set, but the inconsistencies between analyses and animals are just too prevalent.

1. There are many differences between the animals physiologically: I identified a lot of them in the previous review and they are many more that appear now in the new supplementary figures. But perhaps more fundamental for a study trying to understand the physiological underpinning of learning are the data suggesting that the animals learned different things with different dynamics. We can see this in the purely behavioral data of Fig 1 and S1. In Figure 1, Monkey T learns gradually and at an almost linear rote: if you define learning as a change in performance with training, you would say that the learning (the slope) is relatively constant throughout training. By contrast, Monkey X, has a extremely non-linear curve, and clearly the greatest change happened between days 1 and 2. This would be consistent with a pure task familiarity effect: in the first day, he wasn't consistently sure what the task actually was, but as soon as he realized it (by day 2, and possibly in the later half of day 1), his performance improved substantially and had modest improvement after that. This is not really category learning in the sense of refining sensory representations, rather its task learning in trying understand the relationship between sensory representations and reward. Figure S1 supports this: Monkey T increased his sensitivity systematically (the slope of the psychometric), while Monkey X largely did not: he simply changed his criterion and systematically reduced

bias (especially in the first day) without obvious changes in sensitivity. Both bias reduction and sensitivity improvement result in behavior improvement but conceptually (e.g. standard decision theory), and potentially physiologically, are completely different. So we have N=1 for improved category sensitivity, and N=1 for changes in criteria, but not N=2 for either one.

We thank the reviewer for this insightful thought. We agree that the two monkeys display a difference in decision criterion (Figure S1C), probably due to differences in task design (Figure 1A). To quantify the changes in category sensitivity, we fit the psychometric curve each day with a logistic function. We found, for both monkeys, the slopes of the psychometric curves systematically increased across learning days (Figure S1D). Thus, both monkeys showed improved category sensitivity, but Monkey X combined an early rule-discovery/criterion shift with a representational refinement.

We update Figure S1 and figure legend.

Figure S1

Figure S1 (A)Data fit with the logistic function: $a + b / (1 + \exp(-c * (x - d)))$. Fitted curves with an adjusted R^2 below 0.85 (i.e., monkey T's first day) were not displayed..... **(D)** The slope of the psychometric curve ('c' in the above function) increased across learning days for both monkeys. $p = 0.007, 0.03, r = 0.85, 0.91$, for monkey T and X, respectively, Pearson correlation.

We add this result on line 78 – 79: Despite their differing decision biases, both monkeys displayed systematic increases in psychometric slope over training (Figure S1D), indicating enhanced category sensitivity.

We discuss it on line 472 – 478: Both monkeys increased psychometric slope—indicating a shared gain in category sensitivity—but their learning diverged in how the decision boundary evolved: Monkey T’s slope rose steadily with little boundary shift, whereas Monkey X coupled its slope increase to an early, pronounced boundary realignment. Despite this behavioural divergence, both animals exhibited parallel gains in category information within area TE, suggesting that TE plasticity may provide a common representational substrate that can be leveraged by either strategy—rule discovery or sensory sharpening—to support successful visual categorization.

2. Largely based on the previous sets of comments, the authors commendably added a number of analyses, most of which appear in the form of supplementary material. Problematically, I don't find these results very consistent or supportive with the “TE is responsible for learning” claim. Evidence in support of that claim should show that the time course of changes in TE is a significantly better match than changes in TEO to explain both animals' learning dynamics. By simple visual inspection, I would say almost all of the main body figures, with the exception of 3E, are supportive (although I do have statistical concerns which are stated below). However, if I make a check list of supportive data in the supplementary materials that focus on population level analyses, things are a lot less clear. In Figure S3, B is supportive, but A,C, and D are not. Neither S5A or S5B seems supportive. In figure S6 A and C are, but B and D are not. And finally, I don't see anything in S8, S9 or S11 that is particularly supportive.

We appreciate the reviewer’s concern. We carefully re-examined the figures and clarify as follows:

Figure S3 quantified information of morph-level and morph-identity, not category information.

Figure S5 reports choice signals, not category information, so the result does not directly relate to our TE-vs-TEO claim (see responses in the following).

In Figure S6, we now label the panels explicitly: panel A and C (TE) show significant learning-related gains, whereas panel B and D (TEO) do not. Thus, the results are consistent with our claim.

In Figure S8E we found that, relative to TEO, neurons in TE were more likely to exhibit responses dominated by a single category (see Results on line 258 – 269), which supports our claim.

The ROC analyses in S8 F–K show no systematic change for cat-encoding, and we discussed the possible reasons on line 429 – 451.

d' in Fig. S9 (Fig. 3C, F in revised version), mirrors the main-text result: TE increases, TEO does not. Although, on a few days, combining both areas slightly improves decoder performance, this does not contradict our conclusion that TE is the primary locus of category plasticity (also see the responses below).

In Fig. S11 (Fig. S10 in revised version), TE— but not TEO—shows enhanced category information when learning a large cat/dog set; neither area changes for car/truck, likely because that discrimination relies on coarse, low-level features processed earlier in the visual hierarchy (see line 320 – 330).

I'm particularly concerned about the choice analyses: if signals in these regions are not consistently predictive of choices, isn't making the argument that these signals are responsible for better choices dangerous? Doesn't it make the biological relevance of “ideal decoders” less compelling?

We appreciate the reviewer's concern. Trial-by-trial choice predictivity from TE is indeed modest and not uniformly significant (Fig. S5B). Our claim, however, is not that TE alone determines every choice, but that learning raises the ceiling of category information available in TE which could be exploited by downstream circuits. TE activity contains information sufficient to support improved choices but is not the exclusive causal driver of each decision. Downstream areas can integrate it with other signals (e.g., attention, priors, motor noise) to generate the final decision. Nonetheless, our results show that the category information rise in TE tracks both monkeys' behavioral improvement during learning (see “Behavioral performance correlated with neural representations in TE but not TEO”).

We have modified the language in the manuscript. Now there are no expressions like ‘responsible for leaning’, or ‘responsible for choice’.

We update the related discussion on line 391 – 398: **Although category decoding accuracy in TE increased significantly over learning and correlated with behavioral performance, trial-by-trial choice predictability remained modest and unchanged. This dissociation suggests that learning primarily enhances the representation of category structure in TE, while the final decision continues to be shaped by additional variables introduced downstream. Thus, TE provides information sufficient to support improved choices but is not the exclusive causal driver of behavioral output. Downstream areas may integrate the category information from TE with other signals (e.g., attention, priors, motor noise) to generate the final decision.**

3. So if TE isn't solely responsible, is it more responsible than TEO? Here I have to

agree with other reviewer that if A is significant and B is not, the difference isn't necessarily significant. There are three approaches that come to mind here (but I'm sure there are others): one is to specifically look for synergies or redundancies (which the d' does), the other would be to try to do some partial correlation analysis (time course of TE vs learning, given TEO and TEO vs learning given TE), or a regression approach in which the factors and their interactions are all incorporated: $aTE + bTEO + cTE, TEO = \text{learning}$. An additional point here, with regard to synergies/redundancies, is a lost opportunity: if I combine simultaneously recorded TE and TEO neurons into one large population basket, how does it do and how does it compare with TE and TEO separately? Taking the classic signal vs noise correlation approach, if signal correlations between TE and TEO are in a different direction than noise correlations, then you should be able to decode much better with both populations than with either one individually. I think this is what is going on in S9 (which really should be merged with 3B and D in the main text), but why not do this for the rest of the analyses of Figures 3 and 4? And, for the question of learning, does the dynamics of the combined population better match behavioral changes than the dynamics of the individual populations?

We thank the reviewer for the concern about the differences between TE and TEO. We appreciate the novel approaches. In the previous revision, we have tested the differences between TE and TEO throughout the manuscript with a linear regression model that included an interaction term as suggested by reviewer #2.

We have computed d' for TE and TEO ensembles individually and in combination. Combining simultaneously recorded TE + TEO units yields only a modest improvement over TE alone, especially for the generalization category decoder (Figure 3F), which indicates that TE contains more abstract category information than TEO.

We now perform partial correlation analysis on the decoding versus behavior. Partial-correlation analysis showed that TE decoding remained strongly related to behavioral improvement after controlling for TEO and subjects. The reciprocal partial correlation for TEO was not significant.

We added the method on line 697 – 702: Third, we quantified the unique association between TE or TEO population signals and behavior using partial correlations. Specifically, we computed Pearson partial correlations between TE decoding accuracy and behavioral performance while partialling out TEO decoding and monkey identity (0 = monkey T, 1 = monkey X). The reciprocal analysis tested TEO vs. behavior controlling for TE and monkey identity.

We added the results on line 192 – 198: Third, we performed partial-correlation analyses. Across learning, TE decoding remained significantly related to behavior after controlling for TEO and subject (partial $r = 0.87, 0.86, p = 2.73 \times 10^{-4}, 3.25 \times 10^{-4}$ for category decoding and generalization category decoding, respectively). The reciprocal

partial correlation for TEO (controlling for TE and subject) was not significant (partial $r = 0.2, -0.03, p = 0.54, 0.92$ for category decoding and generalization category decoding, respectively). Thus, TE carries unique behavior-relevant variance beyond that recorded from TEO.

I think this is what is going on in S9 (which really should be merged with 3B and D in the main text), but why not do this for the rest of the analyses of Figures 3 and 4?

We merged Figure 3B, D and S9, as well as the related main text (on line 145 – 160) and figure legends.

For the 'rest of the analyses of Figure 3', we computed representational dissimilarities for the pooled TE+TEO population. Across days, the pooled curves were consistently intermediate between TE and TEO for both between-category and within-category correlation distances. We think it does not conflict with the result that pooling TE and TEO yields higher d' (based on decoder scores) on some days. The decoder is supervised for the cat/dog boundary. When we pool TE+TEO, the decoder can re-weight neurons and exploit whatever small, complementary signals in TEO (or TE in early days) to improve discriminability. However, representational dissimilarities $(1-p)$ is an

unsupervised, global measure of representational geometry; it is computed across all pairs of images and weights all neurons equally. Under this averaging, the small, complementary TEO signals (or TE signals in early days) may be diluted. As a result, the pooled correlation distances between-category lie between TE and TEO, even when the supervised decoder shows modest gains. The intermediate pooled curves are also not unexpected for correlation distances within category. Because this analysis does not change our conclusions and adds little new insight, we have not included it in the manuscript.

For Figure 4, we repeated the correct-versus-error analysis with a decoder trained on the concatenated TE + TEO population. The TE + TEO curve tracked TE and remained above TEO, but did not exceed TE on most days (see the following figure, notations were same as Figure 3C&F, red asterisks indicated significantly greater TEO+TE than $\max(\text{TEO}, \text{TE})$, while *n.s.* indicated no significant difference. On the remaining days, TEO+TE was significantly weaker than $\max(\text{TEO}, \text{TE})$). Thus, adding TEO provides little additional information and can even dilute performance occasionally. We have not included the results in the manuscript. We are happy to merge it into a supplement figure set if the reviewer believes it would benefit readers.

And, for the question of learning, does the dynamics of the combined population better match behavioral changes than the dynamics of the individual populations?

Thanks for the reviewer's insightful suggestion. We fit three separate linear models to predict session-by-session behavioral performance during learning: (1) TE d' alone, (2) TEO d' alone, and (3) d' from the combined TE + TEO population. Monkey identity was included as a dummy factor. Information-criterion scores are summarized below:

Predictor	AIC	BIC
Category decoder		
d' (TE)	94.2	96.11

d' (TEO)	108.55	110.47
d' (TE + TEO)	97.16	99.08
Generalization category decoder		
d' (TE)	97.01	98.93
d' (TEO)	107.42	109.33
d' (TE + TEO)	99.01	100.93

The TE-only model yielded the lowest AIC and BIC, indicating it provides the strongest explanation of the behavioral data. Adding TEO to the model actually worsened the fit a little, and the TEO-only model was markedly poorer. Thus, TE population activity alone supports the behavioral improvements, and TEO contributes little additional explanatory power. Further, TEO signals help categorization on some individual days (Figure 3C, F) but don't carry the learning-relevant variance over time.

We added this result on line 185 – 191: Furthermore, pooling TEO with TE did not enhance behavioral prediction when regressing behavior on neural d'. Information-criterion measures, AIC (Akaike Information Criterion) and BIC (Bayesian Information Criterion), were lowest for TE compared with TEO and the combined TE + TEO model (Category decoder: AIC 94.20 (TE), 108.55 (TEO), 97.16 (TE+TEO); BIC 96.11, 110.47, 99.08. Generalization decoder: AIC 97.01, 107.42, 99.01; BIC 98.93, 109.33, 100.93), indicating that TEO contributes little additional learning-relevant variance across days.

We also rephrase the abstract on line 21 – 23: TE and TEO can contribute complementary, partially independent category information. However, TEO does not add learning-relevant variance across days.....